# Integrated epidemiological and molecular data inform the relationship between precancer and cancer states of esophageal adenocarcinoma

**A list of authors and their affiliations appears at the end of the paper**

Cancer generally takes years to evolve, and early diagnosis can prevent life-threatening cancer. Establishing a link between precancerous states and cancer is essential for effective screening and prevention. Esophageal adenocarcinoma (EAC) is an increasingly prevalent, poor-outcome cancer, and its presumed precursor, Barrett's esophagus (BE), characterized by intestinal metaplasia, is evident in only about half of cases. Here to test whether BE is a prerequisite to EAC, we integrated epidemiological and clinical characteristics in a prospective cohort of 3,100 patients with EAC for any evidence of BE (BE-positive and BE-negative) and compared genomic features using a subset of 710 patients with whole-genome sequencing and 87 patients (380 samples) with multiregional whole-exome sequencing. Demographic and genomic features typically associated with BE were observed across BE-positive and BE-negative EAC cases. Notably, molecular features consistent with early BE evolution were detected in both phenotypes. Advanced tumor stage was the only variable that corresponded with increased likelihood of BE-negative EAC, including in some patients with a previous BE diagnosis. Phylogenetic analyses revealed shared evolutionary trajectories, and spatial transcriptomic and proteomic analyses demonstrated intestinal metaplasia-associated lineage markers in both groups. These findings suggest a single pathway to EAC, with implications for early diagnosis and prevention strategies.

Cancer remains a leading cause of morbidity and death globally[1]. Early cancer detection is a priority for healthcare systems worldwide to improve patient outcomes and reduce treatment-related costs[2]. Screening and preventative strategies can have a remarkable impact on cancer incidence and mortality—for example, systematic programs to screen for cervical intraepithelial neoplasia and immunize against human papillomavirus to prevent cervical cancer[3]. When the link between precancers and cancer is unproven or heterogeneous, such as for thyroid nodules, screening and monitoring programs can lead to harm[4,5]. Understanding whether a precancer, as defined by a

pathological precursor state or at-risk tissues, is required for cancer development underpins cancer control strategies[6,7].

EAC is an example of a very-poor-outcome cancer, which has not improved through surveillance schemes for columnar-lined BE[8]. For years, we have assumed that BE is the obligate precursor to EAC, but with the lack of progress in cancer control this is being called into question. Although we have increasing knowledge about EAC, the association between molecular subtypes and risk factors, such as reflux, obesity and tobacco smoking, is poorly delineated[9,10]. Furthermore, studies on the evolution of EAC have, understandably, focused on BE adjacent to the

✉e-mail: shahriar.zamani@nih.gov; rcf29@cam.ac.uk

**Fig. 1 | The study design to establish whether there is a BE-independent pathway to EAC.** The shaded sections show the stages of the study. The prospective cohort comprises 3,100 patients with EAC, and all analyses flow from this cohort. All cases were carefully reviewed to determine their BE phenotype (BE-positive, BE-negative and BE(?) EAC). We ascertained risk factors using questionnaires and prospective case note review at the time of their EAC diagnosis, including demographics, medications, risk factors and clinical information (green shaded panel). The genomics analysis was conducted in all patients with sufficient cellularity from snap-frozen samples, and WGS was favored to capture mutational signatures and structural variants (yellow shaded panel). Because the genomics data on BE are sparser compared to EAC, we also sequenced 388 BE samples to enable a robust comparison (orange to right-hand side of yellow panel). A subset of 380 samples from 87 patients had multiregional FFPE samples—whole exomes were generated and analyzed, enabling us to construct the phylogenetic trees (gray shaded panel). Finally, we scrutinized 214 of 3,100 EAC cases with a BE history prior to their cancer diagnosis and performed spatial transcriptomics and IHC staining in a subset for BE biomarkers, including TFF3 and REG4, in both BE-positive and BE-negative samples. Figure created in BioRender; Wu, L. https://biorender.com/z6york2 (2026).

cancer or samples from different stages of BE in cross-sectional and longitudinal BE cohorts[10–15]. Therefore, we do not have clear data on the distinction or similarity between EAC cases with or without discernable BE.

The question about the importance of BE as a precancerous condition has been bolstered by recent detailed clinico-pathology studies showing that approximately half of EAC cases lack a clinical diagnosis of BE[16,17]. Furthermore, the authors of the landmark population-based, case–control study, which demonstrated that reflux symptoms were strongly associated with EAC (odds ratio: 7.7–43.5, depending on the duration and severity of symptoms), noted that this was the case whether BE was present or not (reported as data not shown), and this has not been investigated further[17].

These observations raise the possibility of an alternative pathway wherein risk factors such as gastroesophageal reflux disease, obesity and smoking induce direct malignant transformation without a metaplastic precursor. However, this hypothesis remains contentious. The absence of BE in these cases may simply reflect challenges in detecting occult or regressed BE or could suggest a fundamentally distinct biological trajectory.

To address this question, we leveraged a prospective EAC cohort study of 3,100 patients that explicitly collected data on the association with BE, along with detailed epidemiological and clinical data and genomic data on a subset of cases. We hypothesized that, if an independent pathway exists, these two groups would be expected to diverge

in their genomic landscapes and associated risk factors. Conversely, extensive overlap would make the dual-pathway hypothesis unlikely and affirm the central role of BE in EAC progression.

Here we report that there are specific demographic and genomic features observed in BE that also correlate strongly with EAC, regardless of the presence of a BE phenotype. Our parsimonious logistic regression model of epidemiological and genomic features was unable to separate the two EAC phenotypes, except for advanced tumor stage in BE-negative EAC. Furthermore, we found that EAC retains transcriptional identity of intestinal metaplasia, supporting a directional model of disease progression. These findings challenge the notion of a BE-independent pathway and underscore the importance of BE, characterized by intestinal metaplasia hallmarks at a genomic and transcriptional level, in EAC carcinogenesis.

## Results

### Study population

The study population included 3,100 patients with EAC undergoing surgical resection prospectively recruited during 2002–2022 from a multicenter consortium, Oesophageal Cancer Clinical and Molecular Stratification (OCCAMS), across 25 UK centers[18]. Ascertainment of the BE status was critical to test our hypothesis. Study sites were trained to include details of a prior history of BE and any evidence of BE on diagnosis. In addition, the endoscopic, surgical and histopathological reports were manually reviewed to classify each case phenotypically according to the presence (BE-positive) or absence (BE-negative) of macroscopic or microscopic evidence of Barrett's intestinal metaplasia or Barrett's dysplasia adjacent to the tumor, and, if it was not possible to be confident, these were denoted BE-unascertainable (BE(?)) and retained for comparison (Extended Data Fig. 1a and Extended Data Table 1a). Specifically, the BE-positive cohort with clear documentation of pathological findings comprised 811 of 1,088 (75%) with intestinal metaplasia documented; of these, a segment length was specified ($n = 357$) or documented as 'amounting to a diagnosis of Barrett's esophagus' ($n = 454$), and the remainder, 277 of 1,088 (25%), had clear endoscopic documentation of >1 cm columnar-lined esophagus. The study design is summarized in Fig. 1 and Extended Data Fig. 1a. All 3,100 cases had comprehensive epidemiological and clinical data collected prospectively, and, of these, a subset of 710 (252 BE-positive, 183 BE-negative and 275 BE(?) EAC) had sufficient cellularity (>70%) for whole-genome sequencing (WGS) with fresh-frozen samples. A subset of EAC cases with multiregional ($n = 380$ samples from 87 patients) whole-exome sequencing (WES) from macro-dissected formalin-fixed, paraffin-embedded (FFPE) tissues and longitudinal data ($n = 214$) enabled us to construct a mutational lineage and decipher the phenotypic history in relation to their cancer diagnosis.

To provide a comprehensive list of genomic factors of BE for comparison with BE-positive and BE-negative EAC, we also characterized a large, novel cohort of cancer-free BE samples (intestinal metaplasia and a measurable segment length >1 cm) that also had sufficient cellularity for WGS ($n = 388$) from fresh-frozen samples (Fig. 1, right-hand orange shaded).

### Epidemiological characteristics are similarly distributed in EAC phenotypes

The epidemiological factors of our prospective cohort were typical for EAC in the UK, including majority age >60 years, male gender and white British ethnicity (Extended Data Table 1b). When comparing risk factors across the phenotypes, we observed that male gender and obese body mass index (BMI) were more likely to be BE-positive EAC, and cigarette smoking was borderline associated with BE-negative cases (Table 1). For BE(?) EAC, the epidemiological data distribution was intermediate between BE-positive and BE-negative cases, suggesting that there was not a bias in those with and without an ascribed phenotype (Extended Data Table 1b). Although the differences in risk factors between the phenotypes looked interesting initially, on further analysis these differences were small in magnitude

and inconsistent; as a result, they did not persist in the fully adjusted model of the complete case analysis and were markedly attenuated in sensitivity analyses (Extended Data Table 2a–c). We were particularly interested in testing the effect of heartburn on the causal pathway because this is a known risk factor for BE. Heartburn was a common symptom (>75%) in both phenotypes although slightly less prevalent in BE-negative cases (odds ratio: 0.56 (95% confidence interval: 0.38–0.83), $P = 0.004$), and no difference was observed compared to BE(?) cases (Extended Data Table 2b,c).

When considering clinical factors, tumor, node, metastasis (TNM) stage was strongly associated with increased likelihood of BE-negative EAC, with an adjusted odds ratio of 2.4 (95% confidence interval: 1.8–3.3, $P < 0.001$) for stage II, 2.9 (95% confidence interval: 2.2–3.9, $P < 0.001$) for stage III and 3.2 (95% confidence interval: 1.7–3.7, $P < 0.001$) for stage IV relative to stage I (Table 1). Results remained robust in sensitivity analyses, and similar findings were found in an additional analysis of BE(?) EAC cases with an unascertainable cancer phenotype (Extended Data Table 2a–c).

### Genome landscape of EAC according to BE phenotype

Next, we compared the genomic landscape of the two EAC phenotypes. We hypothesized that if BE-positive and BE-negative EAC arise through distinct pathways, they would exhibit divergent genomic and early disease features rather than shared BE-associated hallmarks. As the published genome-wide datasets for BE are small, we built a new cohort of 388 BE samples from 256 patients, which would be powered to compare with the sequenced EAC cases (252 BE-positive, 183 BE-negative and 275 BE(?)). The epidemiological risk factors observed in our BE cohort are consistent with the EAC cohort, but they were more likely to be obese (BMI ≥ 25 kg m$^{-2}$), which might be expected due to weight loss experienced in invasive disease, and the vast majority (93%) reported a history of heartburn or were on acid-suppressant medication (Fig. 2a and Extended Data Table 3).

We then identified 21 de novo BE driver genes in the BE cohort using GISTIC2.0 and ratio of non-synonymous mutations to synonymous mutations (d$N$/d$S$) analysis (Extended Data Fig. 1b) and compared their alterations across the cohort. The most commonly altered driver genes among BE samples were *CDKN2A* (48%), *TP53* (27%), *ARID1A* (21%), *SMARCA4* (15%) and *MUC6* (11%) (Fig. 2b). As expected, the frequency of some of these driver gene alterations differed in malignant progression—for example, *TP53* mutation frequency increased dramatically between non-cancerous BE and EAC (27% to >70%). Aside from *TP53*, we also observed events that may be driving progression with an increase in mutation prevalence in EAC, compared to BE, including *SMAD4* (mutation or homozygous loss), *CDK6* (amplification) and *CLDN12* (amplification) and a decrease in *CDKN2A*, *ARID1A* and *SMARCA4*, which may be protective. Most importantly for the primary question we are addressing, the mutational profile of premalignant BE is imprinted in later-stage EAC, and the mutation profile among BE-positive, BE-negative and BE(?) EAC cases was remarkably similar (Fig. 2b and Extended Data Fig. 1d). Notably, *CDKN2A*, one of the earliest mutated genes in BE, has a similar mutation prevalence in both EAC phenotypes (BE-positive: 33% and BE-negative: 33%, $P = 1$).

In addition to gene-level analyses, we performed arm-level copy number analysis to assess broader patterns of genomic instability (Fig. 2c). Significant differences in arm-level copy number alterations were observed between EAC and precancerous BE across all chromosomal arms. However, comparisons between the two EAC phenotypes revealed few differences. The only notable exception was chromosome arm 13q, which showed a borderline higher prevalence of alteration in BE-positive EAC ($P = 0.071$ after false discovery rate (FDR) and Bonferroni correction). Furthermore, genome-wide segmented copy number profiles for the entire cohort are presented in Extended Data Fig. 2a–h, offering a comprehensive view of chromosomal alterations.

The most common base substitution signatures in BE are SBS17a and SBS17b, and the individual as well as combined contributions of

**Table 1 | Association of baseline characteristics with risk of BE-negative EAC compared to BE-positive EAC**

| Characteristic | BE-positive EAC n (%) (n=1,235) | BE-negative EAC n (%) (n=880) | Univariable model OR (95% CI, P) | Fully adjusted model OR (95% CI, P) |
|---|---|---|---|---|
| **Age group at diagnosis** | | | | |
| <50 | 67 (5.4) | 59 (6.7) | 1.00 (referent) | 1.00 (referent) |
| 50–59 | 193 (15.6) | 184 (20.9) | 1.08 (0.72–1.62, P=0.700) | 1.12 (0.74–1.71, P=0.595) |
| 60–69 | 497 (40.2) | 317 (36.0) | 0.72 (0.50–1.06, P=0.094) | 0.72 (0.49–1.07, P=0.108) |
| 70+ | 478 (38.7) | 320 (36.4) | 0.76 (0.52–1.11, P=0.155) | 0.72 (0.48–1.07, P=0.100) |
| **Gender** | | | | |
| Female | 163 (13.2) | 158 (18.0) | 1.00 (referent) | 1.00 (referent) |
| Male | 1072 (86.8) | 722 (82.0) | 0.69 (0.55–0.88, P=0.003) | 0.67 (0.52–0.86, P=0.002) |
| **BMI group at baseline** | | | | |
| Normal | 268 (21.7) | 255 (29.0) | 1.00 (referent) | 1.00 (referent) |
| Overweight | 422 (34.2) | 317 (36.0) | 0.79 (0.63–0.99, P=0.039) | 0.89 (0.70–1.12, P=0.309) |
| Obese | 304 (24.6) | 155 (17.6) | 0.54 (0.41–0.69, P<0.001) | 0.57 (0.44–0.75, P<0.001) |
| Missing | 241 (19.5) | 153 (17.4) | 0.67 (0.51–0.87, P=0.003) | 0.87 (0.63–1.20, P=0.385) |
| **Cigarette smoking status** | | | | |
| Never | 452 (36.6) | 298 (33.9) | 1.00 (referent) | 1.00 (referent) |
| Ever | 665 (53.8) | 532 (60.5) | 1.21 (1.01–1.46, P=0.041) | 1.26 (1.03–1.54, P=0.022) |
| Missing | 118 (9.6) | 50 (5.7) | 0.64 (0.44–0.92, P=0.017) | 0.51 (0.32–0.80, P=0.004) |
| **Aspirin/NSAID use** | | | | |
| Never | 257 (20.8) | 161 (18.3) | 1.00 (referent) | 1.00 (referent) |
| Ever | 328 (26.6) | 243 (27.6) | 1.18 (0.91–1.53, P=0.202) | 1.35 (1.03–1.76, P=0.031) |
| Missing | 650 (52.6) | 476 (54.1) | 1.17 (0.93–1.47, P=0.183) | 1.31 (1.02–1.69, P=0.038) |
| **Heartburn symptoms status** | | | | |
| Absent | 150 (12.1) | 160 (18.2) | 1.00 (referent) | 1.00 (referent) |
| Present | 818 (66.2) | 543 (61.7) | 0.62 (0.49–0.80, P<0.001) | 0.63 (0.49–0.82, P=0.001) |
| Missing | 267 (21.6) | 177 (20.1) | 0.62 (0.46–0.83, P=0.001) | 0.79 (0.56–1.13, P=0.200) |
| **TNM stage** | | | | |
| I | 294 (23.8) | 87 (9.9) | 1.00 (referent) | 1.00 (referent) |
| II | 279 (22.6) | 200 (22.7) | 2.42 (1.80–3.28, P<0.001) | 2.43 (1.79–3.31, P<0.001) |
| III | 481 (38.9) | 447 (50.8) | 3.14 (2.40–4.14, P<0.001) | 2.93 (2.23–3.88, P<0.001) |
| IV | 26 (2.1) | 26 (3.0) | 3.38 (1.86–6.14, P<0.001) | 3.19 (1.74–5.86, P<0.001) |
| Missing | 155 (12.6) | 120 (13.6) | 2.62 (1.87–3.68, P<0.001) | 2.64 (1.88–3.74, P<0.001) |

*Fully adjusted model includes all variables listed in the table: age group at diagnosis, gender, BMI group at baseline, cigarette smoking status, aspirin/NSAID use, heartburn symptoms status and combined TNM stage. Each variable is adjusted for the effects of the others. CI, confidence interval; OR, odds ratio.

these signatures (SBS17) were equally distributed across the two phenotypes (BE-positive EAC: 92% and BE-negative EAC: 90%, P = 0.823) (Fig. 2d). Other mutational signatures were uniformly distributed across both phenotypes, including BE(?) EAC (Extended Data Fig. 2i). No differences were observed in the proportion of samples with whole-genome duplication (BE-positive EAC: 72% versus BE-negative EAC: 76%, P = 0.589), and the total mutational load was similar between the two phenotypes (BE-positive EAC median = 26,425 (interquartile range (IQR): 17,523–6,102) versus BE-negative EAC median = 21,495 (IQR: 13,685–35,913), P = 0.371). We hypothesized a priori that catastrophic events, such as insertion/deletion (indel) rates, aneuploidy rates, extrachromosomal DNA, breakage-fusion-bridge (BFB) events and chromothripsis, could accelerate cancer progression in the BE-negative phenotype, but these events occurred at low frequency and were also equally prevalent in the two phenotypes (Fig. 2d and Extended Data Figs. 2i and 3a–d).

It is also possible that, by restricting the analysis to BE-associated genomic alterations, we missed an alternative genomic pathway in

BE-negative EAC. De novo comparison of a comprehensive list of EAC driver genes[19] from the 252 BE-positive and 183 BE-negative EAC cases again revealed a very similar landscape (Fig. 3a). The median number of driver mutations was 5.0 in BE-positive cases (IQR: 3.0–8.0) and 5.0 in BE-negative EAC cases (IQR: 3.0–7.0) (P = 0.871). Of the 76 driver genes known to be associated with EAC[19], differences were observed for four genes between the two phenotypes: MDM2, an inhibitor of p53 dysregulation (P = 0.003), MBD6 (P = 0.004), TP53 (P = 0.040) and SCN3A (P = 0.049); however, these differences were no longer statistically significant after applying Bonferroni or FDR correction for multiple comparisons. Multiple correspondence analysis (MCA) revealed considerable overlap in the distribution of tumors mutated for the 76 EAC driver genes between the two phenotypes (Fig. 3b). Consistent with the full cohort epidemiological analysis, comparison between sequenced cases showed that the BE-negative phenotype is significantly more advanced (TNM (P < 0.001)), and T stage is the most relevant for esophageal phenotype (P < 0.001) (Extended Data Fig. 3e).

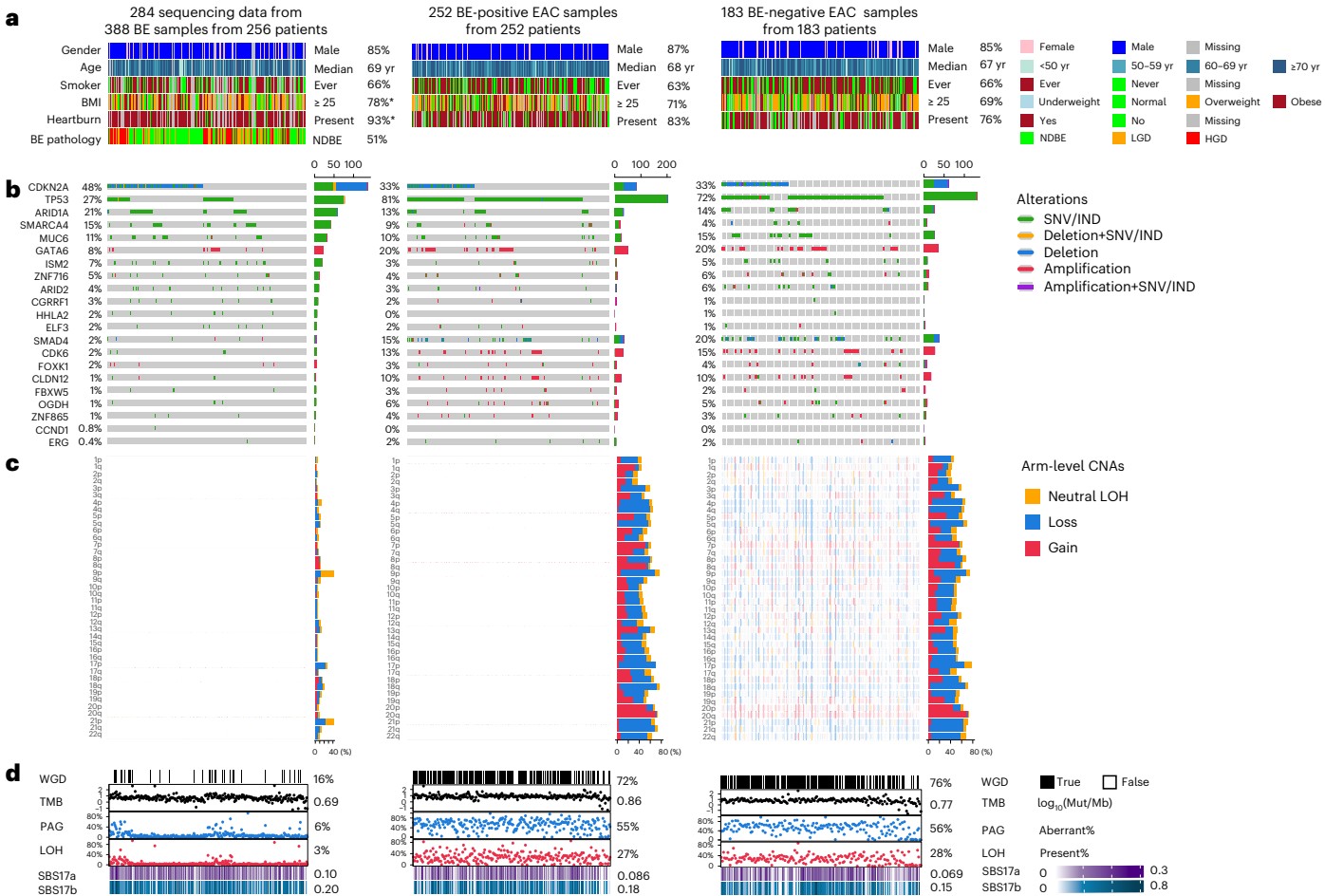

**Fig. 2 | The comparison of high-risk epidemiological factors and genomic features of BE driver genes across BE cohort and different EAC phenotypes.**
**a**, Comparison of epidemiological factors: gender, age, smoking status, BMI and heartburn history across cancer-free BE, BE-negative and BE-positive EAC phenotypes. Cases across both EAC phenotypes had BE risk factors. The BE cohort exhibited a higher percentage of overweight patients compared to the EAC cohorts ($P = 0.018$), likely reflecting the influence of dysphagia, disease burden and EAC treatment on body weight. All phenotypes had a strong association with heartburn, although this was most commonly reported in BE ($P < 0.001$). $\chi^2$ analysis (two-sided) with Bonferroni corrections for multiple comparisons was applied. **b**, Genetic landscape of BE driver genes across cohorts. As expected, there are differences in EAC compared to precancerous BE, (Bonferroni-adjusted $P < 0.001$), but the landscape was remarkably similar in both EAC phenotypes. *CDKN2A*, one of the earliest mutated genes in BE, has a similar mutation prevalence in both EAC phenotypes (BE-negative EAC: 21% and BE-positive EAC: 23%, $P = 0.751$). A marginal difference in *TP53* between the two EAC phenotypes was noted ($P = 0.04$, two-sided $\chi^2$ test), which was lost after Bonferroni adjustment ($P = 0.81$). The comparative analysis results of the genes can be found in Source Data Fig. 2. SNV, single nucleotide variants; IND, indel. **c**, Arm-level copy number

profiles across cohorts. Significant differences in arm-level copy number alterations were observed between EAC and precancerous BE across multiple chromosomal arms. By contrast, comparisons between the two EAC phenotypes revealed few differences. The only notable exception was chromosome arm 13q, which showed a borderline higher prevalence of alteration in BE-positive EAC ($P = 0.071$ after FDR and Bonferroni correction, two-sided $\chi^2$ analysis). The comparative analysis results across all arms can be found in Source Data Fig. 2. **d**, Evaluation of genome-wide events in EAC development: whole-genome doubling (WGD), tumor mutational burden (TMB), percentage of aberrant genome (PAG), loss of heterozygosity (LOH), signature of base substitution 17a (SBS17a) and signature of base substitution 17b (SBS17b). As expected, WGD, TMB, PGA and LOH are significantly lower in BE (Bonferroni-adjusted $P < 0.001$, two-sided $\chi^2$ analysis) than the EAC samples, whereas SBS17a and SBS17b show a higher contribution in the BE cohort, with adjusted $P < 0.001$ and $P < 0.006$, respectively. Of note, the two EAC phenotypes consistently display similar genome-wide patterns, including presence of SBS17. There is a marginal difference in TMB ($P = 0.02$) that does not remain statistically significant after FDR or Bonferroni correction. The values are presented as median. The SBS values are proportions of the signature in samples. CNA, copy number alteration; Mut/Mb, mutations per megabase; yr, years.

---

Given that EAC phenotypes were classified based on both endoscopic (macro) and pathological (micro) evidence of BE, with some cases missing one label (Extended Data Table 1a), we conducted four subgroup comparisons to explore if the findings were consistent across different BE classification criteria: (1) both macro and micro positive versus both negative (most stringent); (2) micro positive versus fully negative; (3) micro positive with versus without macro confirmation; and (4) macro positive with versus without micro confirmation. Across all comparisons, we observed no significant differences after statistical correction in epigenetic profiles, BE-related genomic features (Supplementary Fig. 1) or EAC driver gene alterations (Supplementary Fig. 2).

## Logistic regression model to distinguish BE-negative and BE-positive EAC

Although we observed many similarities between the two phenotypes, there were some subtle differences. Therefore, we developed a logistic regression model for the 435 (183 BE-negative and 252 BE-positive EAC) patients with complete epidemiological, clinical and WGS data to ascertain which features differentiated between the two phenotypes, including key genomic features of BE (CDKN2A and SBS17) and the most relevant clinical factors (T stage). In the unadjusted model, none of these features distinguished between the two phenotypes apart from clinical T stage. Furthermore, when adjusted for age at diagnosis, gender, cigarette smoking status, BMI and heartburn status, T stage remained

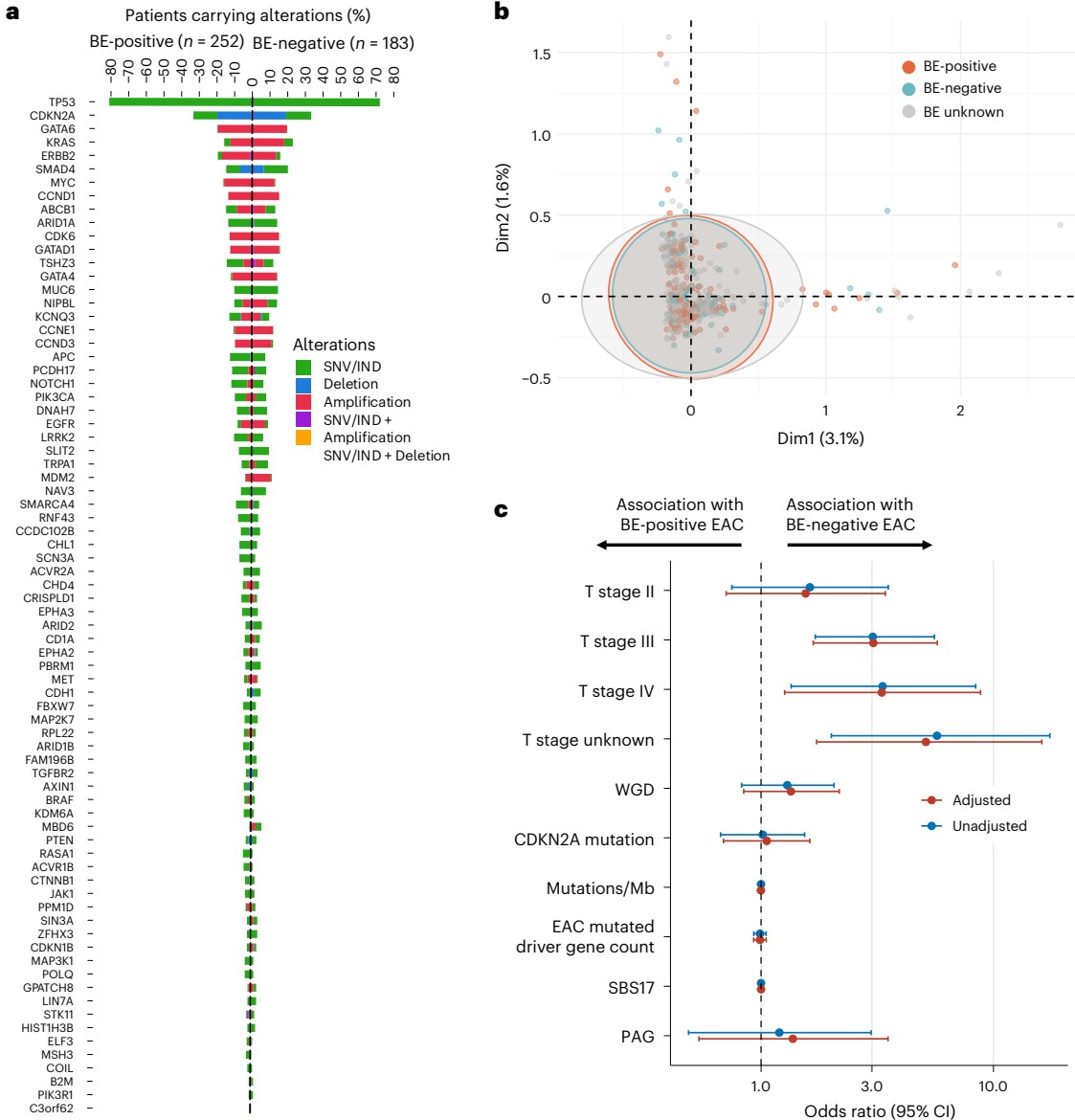

**Fig. 3 | Comparative analysis of genomic variations and stage differences in EAC between BE-positive and BE-negative phenotypes. a**, The prevalence of genomic variations in 76 previously reported EAC driver genes between BE-positive and BE-negative EAC phenotypes. The genes are ranked by the overall prevalence of both phenotypes and grouped by BE-positive and BE-negative. Four genes showed differences between the two groups using a two-sided $\chi^2$ test without multiple comparisons, including *MDM2* ($P = 0.003$), *MBD6* ($P = 0.004$), *TP53* ($P = 0.040$) and *SCN3A* ($P = 0.049$); however, these differences disappeared after applying Bonferroni adjustment for multiple comparisons. SNV, single nucleotide variant; IND, indel. **b**, MCA of EAC phenotypes. This analysis uses MCA to explore the genomic pattern between different EAC phenotypes based on the variations of the 76 identified EAC driver genes. MCA quantifies and displays the genomic patterns by transforming the variation data into principal components and plotting these components on a two-dimensional graph. Each dot on the plot represents a sample; dots that are closer together indicate greater similarity

in genomic patterns. Confidence ellipses around the points for each category visualize the variability and clustering of samples within each phenotype. The overlap of these ellipses across phenotypes suggests that their genomic landscapes are notably similar. **c**, Logistic regression analysis of epidemiological and genomic features between two esophageal cancer phenotypes. A logistic regression model was developed for all 435 (252 BE-positive and 183 BE-negative EAC) patients with complete epidemiological, clinical and WGS data to understand whether any features differentiated between the two phenotypes. Odds ratio greater than 1 with a non-null-overlapping 95% confidence interval indicates association with BE-negative EAC phenotype, and odds ratio less than 1 indicates association with BE-positive EAC phenotype. Adjustments include age at diagnosis, gender, cigarette smoking status, BMI and heartburn status. The measure of center for the error bars shows 95% confidence interval. CI, confidence interval; Mb, megabase; PAG, percentage of aberrant genome; WGD, whole-genome doubling.

the only predictor that statistically significantly increased the odds of BE-negative EAC, independent of the length of the tumor (Fig. 3c).

## Multiregional mutational lineage tracing and clinical surveillance of patients with BE

To better understand the genomic evolution of tumors in BE-negative compared to BE-positive EAC groups, we generated multiregional

WES data (380 samples from 58 BE-positive and 29 BE-negative cases) (Extended Data Fig. 4a). This enabled us to build phylogenetic trees for each case (Source Data Fig. 4a) and cluster them based on evolutionary trajectory (Fig. 4a).

Most EAC cases were initiated by *TP53* mutations, whereas a subset followed alternative evolutionary paths involving genes such as *ARID1A* and *PCDH17*, giving a total of eight phylogenetic clusters. Patients with

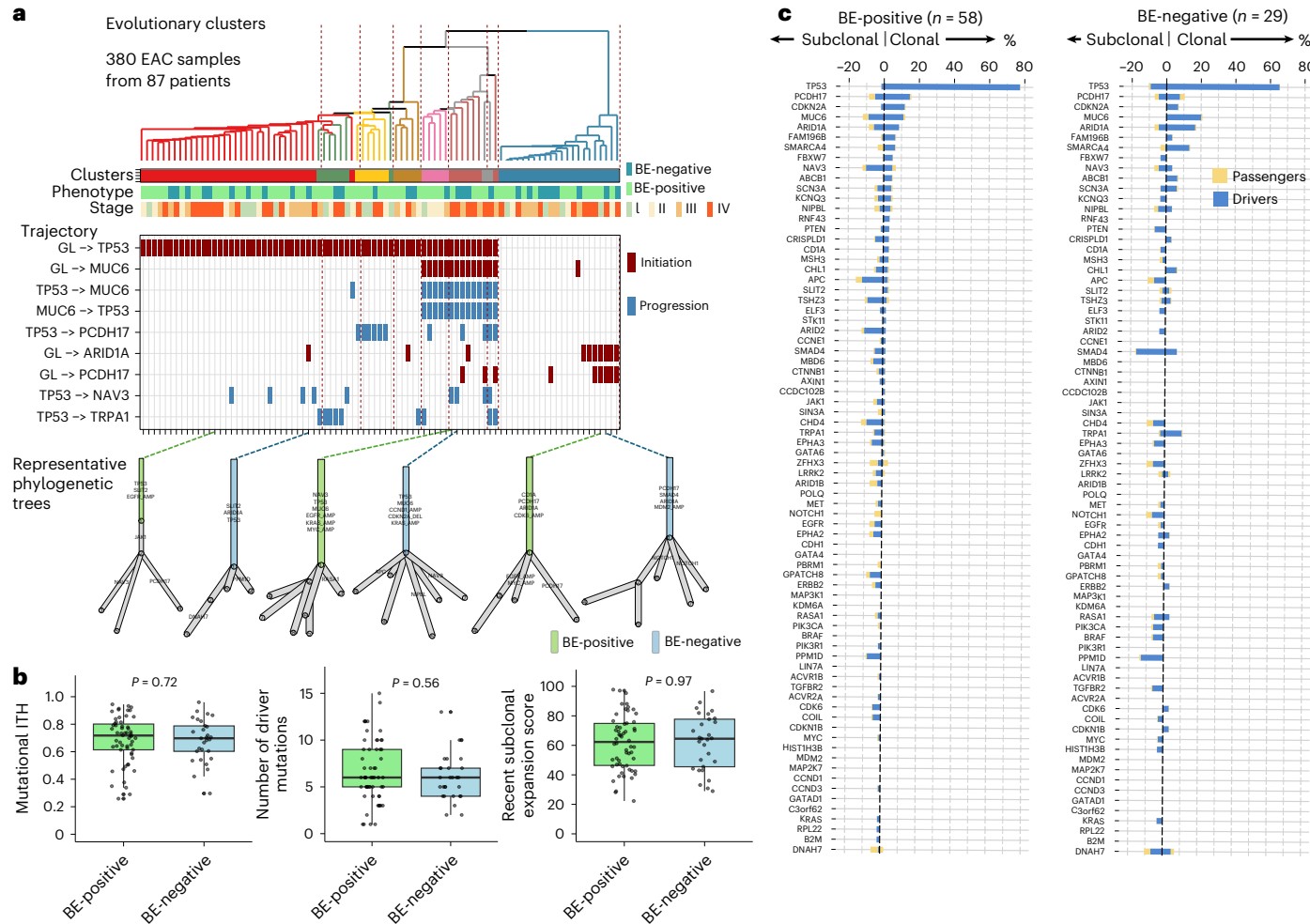

**Fig. 4 | Phylogenetic analysis of evolutionary trajectories and driver gene clonality in EAC. a**, Evolutionary trajectories and clustering across the cohort. We analyzed all patients with multiregional sampling using the REVOLVER algorithm. Most EAC cases were initiated by *TP53* mutations, whereas a subset followed alternative evolutionary paths involving genes such as *ARID1A* and *PCDH17*. Patients with BE-positive and BE-negative phenotypes were randomly distributed across trajectory clusters, and phylogenetic trees within the same cluster showed similar structures regardless of BE status. The trajectory list was filtered to include only those with occurrences greater than eight; the full list of phylogenetic trees is available in the Source Data, and the full list of evolutionary trajectories and driver genes is shown in Extended Data Fig. 4a. A total of 400 samples from 93 patients were sequenced in the whole exosome cohort. After excluding 13 samples from four patients with hypermutation and seven samples from two patients that failed copy number quality control, 380 EAC samples from 87 patients were included in the evolutionary trajectory analysis

after expanding this analysis to the full WES cohort (Extended Data Fig. 4b). **b**, Comparative analysis of phylogenetic features across the cohort (BE-positive, *n* = 58 and BE-negative, *n* = 29). Metrics including intratumour heterogeneity, recent subclonal expansion score, number of driver mutations and others were compared between BE-positive and BE-negative EACs. No statistically significant differences were observed. Additional metrics are provided in Extended Data Fig. 4c. Box plots are presented as median and quartile range. Two-sided Wilcoxon test was performed without adjustment for multiple comparisons. **c**, Clonality of driver gene mutations across groups. *TP53* mutations were predominantly clonal both in patients with BE-positive (97.9%) and patients with BE-negative (87.5%), similar to early-event genes such as *CDKN2A*. By contrast, mutations in genes such as APC, SMAD4 and CHD4 were more frequently subclonal. Comparison of the number of patients with clonal versus subclonal mutations for each driver gene revealed no statistically significant differences between BE-positive and BE-negative EACs.

BE-positive and BE-negative phenotypes were randomly distributed across these trajectory clusters, and, hence, phylogenetic trees within the same cluster showed similar structures regardless of BE status (Fig. 4a, Extended Data Fig. 4b,c and Source Data Fig. 4a,c). Metrics including intratumour heterogeneity, recent subclonal expansion score, number of driver mutations and others were similar between BE-positive and BE-negative EACs (Fig. 4b and Extended Data Fig. 4c). We also compared the proportion of clonal to subclonal mutations according to the BE phenotype. TP53 mutations were predominantly clonal both in patients with BE-positive and patients with BE-negative (87.5%), similar to early-event genes such as CDKN2A. By contrast, mutations in genes such as APC, SMAD4 and CHD4 were more frequently subclonal. Comparison of the number of patients with clonal versus subclonal

mutations for each driver gene revealed no statistically significant differences between BE-positive and BE-negative EAC (Fig. 4c).

## Clinical surveillance patterns and BE lineage biomarker spatial analysis

To clarify whether BE is the necessary precursor, we examined patients with a previous diagnosis of BE. We could substantiate 214 (5% of the cohort) who underwent BE surveillance before their primary diagnosis, and, of these, 166 had a clear BE phenotype (BE-positive or BE-negative and not BE(?)) at EAC diagnosis; this small group with prior BE is in keeping with the de novo symptomatic presentation of EAC. As expected, this subgroup comprised higher proportions of BE-positive (surveillance cohort versus full cohort: 87.3% (145/166)

versus 58.4% (1,235/2,115) were BE-positive, $P < 0.001$) and T1 stage tumor (surveillance cohort versus full cohort in patients with recorded T stage: 56.0% (102/182) versus 19.3% (528/2,740), $P < 0.001$), suggesting some benefit from surveillance (Fig. 5a, Extended Data Table 4 and Supplementary Table 1). Notably, 12.7% (21/166) of cases with known prior BE had BE-negative EAC. Median time from the start of surveillance to an EAC diagnosis was 114 months in BE-negative EAC (IQR: 54–127) and 49 months (IQR: 22–98) in BE-positive EAC. These data suggest that BE-negative patients with EAC in surveillance had less frequent or less effective surveillance or that they had been lost to follow-up. To avoid potential bias introduced by this surveillance cohort to the full cohort in the regression analysis presented in Fig. 3, we performed a sensitivity analysis excluding these 214 patients. The similarity between BE-positive and BE-negative EAC phenotypes remained consistent (Extended Data Table 2a).

We next turned to expression profiling to understand whether there is a transcriptional legacy of BE markers within EAC tissues. We handpicked seven samples (five EAC and two BE) from six patients with EAC with high-quality fresh-frozen tissues (Extended Data Fig. 5a) and performed spatial transcriptomic profiling using Visium HD WT (approximately 18,000 genes) (Fig. 5b) at a 2-μm bin size resolution. Of the five EAC samples, three had a BE-negative phenotype and, thus, lacked adjacent BE.

Unsupervised clustering revealed distinct transcriptomic regions. In EAC samples, one prominent cluster showed significant expression of tumor-associated genes (ERBB2, CLDN4, SPINK1, MKI67, EPCAM and TOP2A) and proliferation pathways, covering a large portion of tumor epithelium. Interestingly, BE lineage genes, including CDX2, REG1A, PIGR, DMBT1, LCN2, CEACAM5, CEACAM6 and OLFM4, were also expressed in this cluster (Extended Data Fig. 5b). In BE samples, the transcriptional landscape was more compartmentalized. One major cluster (Extended Data Fig. 5c, cluster 1, blue) aligned with Barrett's epithelium based on pathology and gene markers, with strong expression of MUC2, AGR2 and REG4, consistent with goblet cell differentiation. Another cluster (cluster 3, green) displayed a tumor-like transcriptomic profile resembling EAC and corresponded to high-grade dysplasia (HGD) region on pathology review. Interestingly, in all cases, BE markers were expressed in tumor regions, whereas EAC-associated genes were not transcribed in non-dysplastic BE regions. The raw images of spatial transcriptomics data are presented in the Source Data.

To further confirm the distinction between BE and EAC profiles, we restricted the analysis to highly specific BE lineage markers to label histologically distinct squamous, stromal, BE and EAC regions. The regions were determined using marker genes curated from the literature (Extended Data Table 5). This confirmed that the BE transcriptome

profile, determined from benign BE tissue, overlaps spatially with the EAC transcriptome map even in EAC cases without adjacent BE, with an average of 56 ± 12% (mean ± s.e.m.) of the tumor tissue expressing BE lineage markers at cellular-scale resolution, with some samples having as high as 90% co-localization (Fig. 5b). On the other hand, the EAC transcriptome was not expressed in BE except for regions of HGD. There was no overlap with squamous and stromal genes, as expected.

Next, we assessed whether BE lineage markers identified by spatial analysis—trefoil factor 3 (TFF3) and regenerating family member 4 (REG4)—were retained at the protein level in BE-negative EAC using immunohistochemistry (IHC) (Fig. 5c). REG4 or TFF3 expression was detected in 50% (4/8) of BE-negative EACs and in all (10/10) of BE-positive EACs, all in moderate or well-differentiated regions. In BE-negative EACs, staining appeared in cytoplasmic vacuoles of glandular epithelium infiltrated by tumor, whereas BE-positive EACs showed strong expression in both tumor regions and adjacent intestinal metaplasia. Notably, all marker-positive BE-negative EACs were in areas that retained some level of cell differentiation: BE-positive EACs (one well, four moderate and five moderate to poor differentiation status) largely showed strong staining, with two moderate to poorly differentiated samples showing focal staining, whereas BE-negative EACs (one moderate, three moderate to poor and four poor) only expressed markers in moderate or moderate to poor tumors, with no staining in poorly differentiated samples. These findings suggest that BE-associated features are retained in EACs, which retain some degree of cell differentiation even in those lacking visible BE at diagnosis.

## Discussion

Assembly of a large, prospective cohort of 3,100 EAC cases with detailed endoscopic, surgical and histopathological data allowed us to ascertain the presence or absence of discernable BE on a case-by-case basis to address similarities and differences in epidemiological, clinical and genomic characteristics. Despite us setting out to identify a BE-independent pathway, we found that the epidemiology and genomic features of BE-positive and BE-negative EAC are remarkably similar. The only feature that robustly separates the two phenotypes is the tumor stage, whereby BE-negative EAC cases are more advanced. Overall, the extensive overlap in the epidemiology and genomic characteristics makes a dual-pathway hypothesis for EAC unlikely and affirms the central role of BE in EAC progression. In keeping with the notion that the cancer can overgrow the BE segment, BE-negative EAC cases were observed in some individuals known to have BE before their cancer diagnosis, and BE intestinal metaplasia lineage markers were found to be translated and expressed in both BE-positive and BE-negative EAC.

**Fig. 5 | Clinical surveillance patterns and BE lineage biomarker spatial analysis in EAC samples. a**, Distribution of T stage and BE status in the surveillance versus full cohort and surveillance durations for patients with prior BE diagnosis. The proportions of early-stage cancers (T1) and BE-positive EAC were higher in the surveillance cohort (214 patients) than in the full cohort (3,100 patients), as expected. When excluding patients with unknown BE phenotypes, the proportions of patients with BE-positive EAC are 87.3% (145/166) in the surveillance cohort and 58.4% (1,235/2,115) in the full cohort. '*' indicates a statistically significant difference in T stage distribution between the surveillance cohort and the full cohort within the same phenotype. '~' indicates a statistically significant difference in T stage distribution compared to BE-positive EAC within the same cohort. $\chi^2$ analysis with FDR and Bonferroni corrections for multiple comparisons was applied. Exact $P$ values for each pair of comparisons are presented in Supplementary Table 1. Among 214 patients with a self-reported prior BE diagnosis, recorded surveillance duration was available for 116 patients, and the median surveillance time was longer in patients with BE-negative EAC (114.0 months (IQR: 54.0–127.0)) compared to patients with BE-positive EAC (49.2 months (IQR: 21.6–97.9)). **b**, Spatial transcriptomic analysis using the 10x Genomics Visium HD platform. EAC regions express BE marker genes, whereas areas of intestinal metaplasia do not express EAC-associated

genes, supporting a unidirectional progression from BE to EAC. Column 1 displays expression overlays for four gene sets. Tumor genes: MKI67, SPINK1, ERBB2 and CLDN4 (dark blue); BE genes: MUC2, TFF3, REG4 and CDX2 (light blue); squamous genes: DSG3, KRT5, KRT14 and TP63 (dark green); and stroma genes: DSG3, KRT5, KRT14 and TP63 (light green). Column 2 highlights spatial expression of the key established protein markers for intestinal metaplasia TFF3 and REG4. **c**, IHC staining of BE-associated biomarkers (TFF3 and REG4) from BE-positive and BE-negative phenotype EAC. In BE-negative EAC cases, positive staining was observed in cytoplasmic vacuoles of glandular epithelium infiltrated by tumor cells in areas where differentiation was maintained. By contrast, BE-positive EACs showed strong expression of both markers within regions of intestinal metaplasia adjacent to the tumor. Note that the cases shown in **b** and **c** are different; only the group labels are consistent across both. **d**, Summary graphic to show how the clinical and pathological measurable evidence of BE varies according to disease stage once an adenocarcinoma is developing. The molecular 'signatures' or characteristics of the BE metaplasia from which the EAC arises are persistent over time. These signatures include REG4 and TFF3 lineage markers ascertainable at the transcriptomic or protein level, and SBS17 and loss of CDKN2A hallmarks retained in the tissue, but this list in not exclusive.

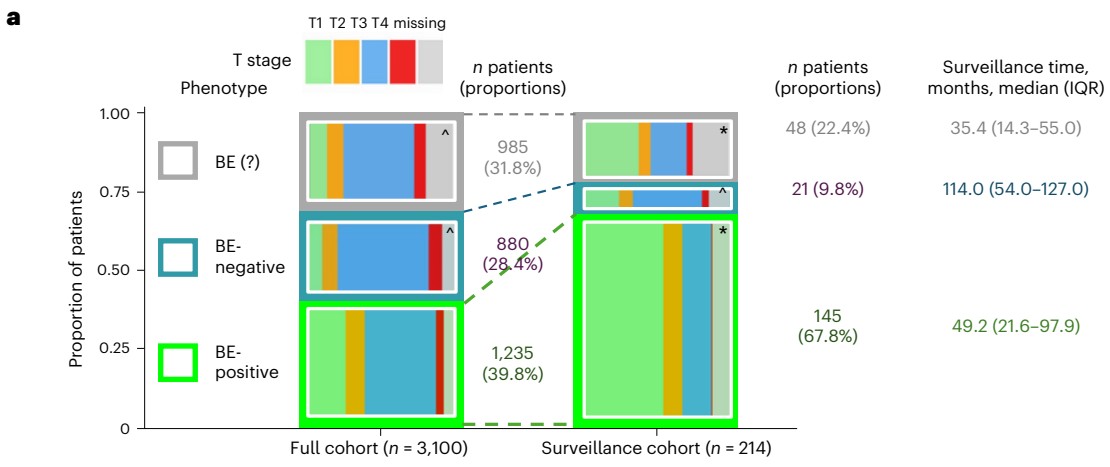

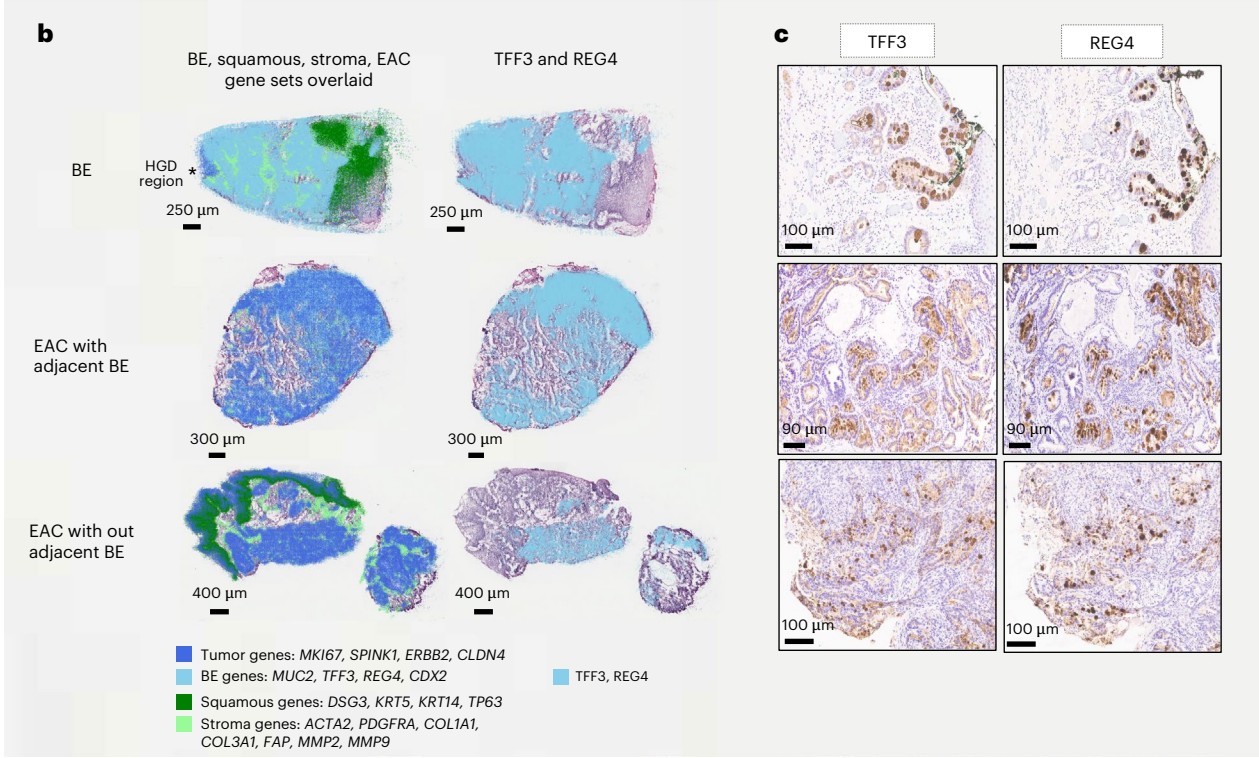

Tumor genes: *MKI67, SPINK1, ERBB2, CLDN4*
BE genes: *MUC2, TFF3, REG4, CDX2*
Squamous genes: *DSG3, KRT5, KRT14, TP63*
Stroma genes: *ACTA2, PDGFRA, COL1A1, COL3A1, FAP, MMP2, MMP9*
TFF3, REG4

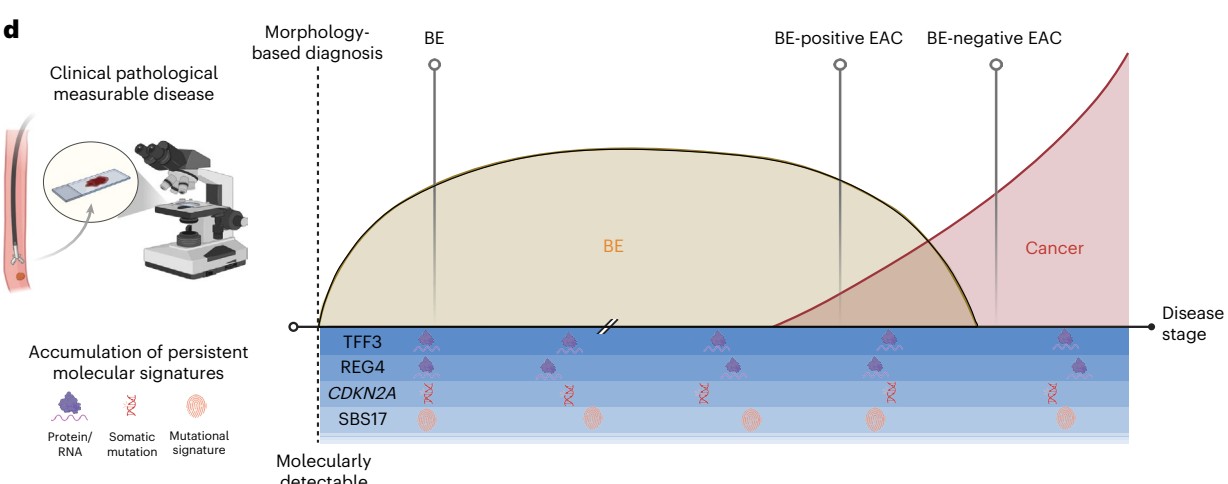

These findings are consistent with our previous single-cell RNA sequencing data from a much smaller cohort that could not distinguish between EAC phenotypes based on their transcriptomic profile[20]. In addition, a multiscale computational model for the transition probabilities from normal esophagus through BE to EAC recapitulates the incidence of EAC from observed US population data, suggesting that BE is the likely precursor for all EAC[21].

From a genomic perspective, one might expect to find clues for whether BE-negative EAC could harbor features that lead to a more aggressive born-to-be-bad phenotype that could also explain the poorer prognosis previously observed in BE-negative EAC cases, independent of stage[4,22,23]. Catastrophic alterations such as BFB cycles and extrachromosomal DNA (ecDNA) are present in approximately 25–30% of tumors and have been associated with poor prognosis[24,25], and a model of catastrophe-driven EAC development with rapid progression from BE has been proposed[26]. However, these events occurred rarely and at a similar rate in the two phenotypes.

The transcriptomic and protein data suggest a unidirectional program from BE to EAC, in which EAC retains, rather than erase, the molecular characteristics of its precursor state. The prominent lineage markers, such as TFF3 and REG4, could act as a 'smoking gun'. Indeed, the large-scale plasma proteomic analysis from the UK Biobank showed that REG4 was associated with an increased risk of EAC (2.02 (95% confidence interval: 1.66–2.45))[27], and these secreted proteins have been observed in other gastrointestinal adenocarcinomas. Furthermore, short and even ultra-short segments of BE, characterized by intestinal metaplasia without a measurable columnar-lined segment, which are most likely to be overgrown by cancer at the time of a symptomatic diagnosis, could account for a substantial proportion of EAC, which we currently deprioritize due to the burden that it could place on our endoscopic services[28,29]. A shift toward a quantitative, minimally invasive readout to find individuals at risk could provide a feasible route to impact the high mortality associated with late diagnosis of EAC. However, additional molecular risk stratification beyond evidence of intestinal metaplasia would be critical to prevent overdiagnosis.

The strengths of our study lie in the well-powered cohort that was prospectively collected with very detailed phenotyping data, along with epidemiological and matched WGS in a substantial subset that was combined for the logistic regression analysis. The study also has limitations. There are cases where we could not ascertain the BE status, and these are, therefore, included as an indeterminate category. The cohort of patients who had previous BE and were under surveillance provides an informative longitudinal dataset, but the clinical details, such as the follow-up time and intervals, were incomplete because these were real-world data and endoscopy records are not always comprehensive. Unlike some other cancers, such as pancreatic neoplasia[30], we cannot conceive of an experimental approach to address this question, because animal models differ fundamentally from humans with respect to the squamous-lined forestomach, the gastroesophageal junction anatomy and lack of reflux, because they are quadrupedal. Furthermore, the clinical observation that 12.7% (21/166) of EAC can overgrow known BE to oblivion in the surveillance subset lends further support to our interpretation. We would have preferred to expand and confirm our results in an independent cohort. This cohort is UK based, which is highly relevant for this cancer type that has the highest EAC incidence and associated mortality worldwide, and, unfortunately, other EAC datasets (for example, The Cancer Genome Atlas (TCGA)) do not include detailed phenotypic evaluation to allow us to study this question about the association with BE.

In conclusion, intestinal metaplasia, which may or may not amount to a columnar-lined segment >1 cm, can be considered the precursor lesion to EAC. The evidence of this precursor lesion can be obscured at diagnosis in more advanced cases. Although this finding may not be surprising, without evidence to the contrary it has remained a clinically important and intriguing question dogging the field. Our results lend credence to efforts to improve cancer control through screening to identify individuals with transcriptional and genomic hallmarks of intestinal metaplasia who are at risk, although whether this would reduce population mortality remains to be determined.

The integrated large-scale epidemiological and genomic approach used here could be informative for deciphering from human data whether other putative precancerous lesions, such as lung or thyroid nodules, or pathological entities, such as pancreatic intraepithelial neoplasia or serous tubal intraepithelial carcinoma lesions in the fallopian tubes, are required steps in cancer evolution. Understanding the necessary and sufficient steps for cancer evolution is important to determine optimum approaches for earlier detection of this globally important disease.

## Online content

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

**Shahriar A. Zamani** [1,2,36,39] ✉, **Lianlian Wu**[1,39], **Emily L. Black**[1], **Alexander Bartram** [1], **Alvin W. T. Ng** [1,3,37], **Maria Secrier** [4], **Jacqueline D. Perelman**[1], **Ahsen Ustaoglu** [1], **Emma Ococks**[1], **Daniel Jacobson**[1], **Ginny Devonshire**[1], **Nicola Grehan**[1], **Barbara Nützinger**[1], **Adam Freeman** [1], **Ahmad Miremadi**[5], **Maria O'Donovan**[5], **Alexander M. Frankell** [1], **Sarah Killcoyne** [1,38], **Oesophageal Cancer Clinical and Molecular Stratification (OCCAMS) Consortium**\*, **Helen G. Coleman**[6] & **Rebecca C. Fitzgerald** [1] ✉

[1]Early Cancer Institute, Department of Oncology, University of Cambridge, Cambridge, UK. [2]Cancer Prevention Fellowship Program, Division of Cancer Prevention, National Cancer Institute, National Institutes of Health, Rockville, MD, USA. [3]Cancer Research UK Cambridge Institute, University of Cambridge, Cambridge, UK. [4]UCL Genetics Institute, Department of Genetics, Evolution and Environment, University College London, London, UK. [5]Department of Pathology, Cambridge University Hospital NHS Trust, Cambridge, UK. [6]Cancer Epidemiology Research Group, Centre for Public Health, Queen's University Belfast, Belfast, UK. [36]Present address: Division of Cancer Epidemiology & Genetics, National Cancer Institute, National Institutes of Health, Rockville, MD, USA. [37]Present address: Lee Kong Chian School of Medicine, Nanyang Technological University, Singapore, Singapore. [38]Present address: Cyted Ltd., Cambridge, UK. [39]These authors contributed equally: Shahriar A. Zamani, Lianlian Wu. \*A list of authors and their affiliations appears at the end of the paper. ✉e-mail: shahriar.zamani@nih.gov; rcf29@cam.ac.uk

### Oesophageal Cancer Clinical and Molecular Stratification (OCCAMS) Consortium

**Rebecca C. Fitzgerald**[1], **Paul A. W. Edwards**[1,3], **Nicola Grehan**[1,7], **Barbara Nutzinger**[1], **Aisling M. Redmond**[1], **Christine Loreno**[1], **Sujath Abbas**[1], **Adam Freeman**[1], **Elizabeth C. Smyth**[7], **Maria O'Donovan**[1,5], **Ahmad Miremadi**[1,5], **Shalini Malhotra**[1,5], **Monika Tripathi**[1,5], **Calvin Cheah**[1], **Hannah Coles**[1], **Curtis Millington**[1], **Matthew Eldridge**[3], **Maria Secrier**[3], **Ginny Devonshire**[1], **Sriganesh Jammula**[3], **Jim Davies**[8], **Charles Crichton**[8], **Nick Carroll**[7], **Richard H. Hardwick**[7], **Peter Safranek**[7], **Andrew Hindmarsh**[7], **Vijayendran Sujendran**[7], **Stephen J. Hayes**[9,10], **Yeng Ang**[9,11,12], **Andrew Sharrocks**[12], **Shaun R. Preston**[13], **Izhar Bagwan**[13], **Vicki Save**[14], **Richard J. E. Skipworth**[14], **Ted R. Hupp**[15], **J. Robert O'Neill**[7,14,15], **Olga Tucker**[16,17], **Andrew Beggs**[16,18], **Philippe Taniere**[16], **Sonia Puig**[16], **Gianmarco Contino**[16], **Timothy J. Underwood**[19,20], **Robert C. Walker**[19,20], **Ben L. Grace**[19], **Jesper Lagergren**[21,22], **James Gossage**[21,23], **Andrew Davies**[21,22], **Fuju Chang**[21,23], **Ula Mahadeva**[21], **Vicky Goh**[23], **Francesca D. Ciccarelli**[23], **Grant Sanders**[24], **Richard Berrisford**[24], **David Chan**[24], **Ed Cheong**[25], **Bhaskar Kumar**[25], **L. Sreedharan**[25], **Simon L. Parsons**[26],

**Irshad Soomro[26], Philip Kaye[26], John Saunders[9,26], Laurence Lovat[27], Rehan Haidry[27], Michael Scott[28], Sharmila Sothi[29], Suzy Lishman[3,30], George B. Hanna[31], Christopher J. Peters[31], Krishna Moorthy[31], Anna Grabowska[32], Richard Turkington[33], Damian McManus[33], Helen Coleman[33], Russell D. Petty[34] & Freddie Bartlett[35]**

[7]Cambridge University Hospitals NHS Foundation Trust, Cambridge, UK. [8]Department of Computer Science, University of Oxford, Oxford, UK. [9]Salford Royal NHS Foundation Trust, Salford, UK. [10]Faculty of Medical and Human Sciences, University of Manchester, Manchester, UK. [11]Wrightington, Wigan and Leigh NHS Foundation Trust, Manchester, UK. [12]GI Science Centre, University of Manchester, Manchester, UK. [13]Royal Surrey County Hospital NHS Foundation Trust, Guildford, UK. [14]Edinburgh Royal Infirmary, Edinburgh, UK. [15]Edinburgh University, Edinburgh, UK. [16]University Hospitals Birmingham NHS Foundation Trust, Birmingham, UK. [17]Heart of England NHS Foundation Trust, Birmingham, UK. [18]Institute of Cancer and Genomic sciences, University of Birmingham, Birmingham, UK. [19]University Hospital Southampton NHS Foundation Trust, Southampton, UK. [20]Cancer Sciences Division, University of Southampton, Southampton, UK. [21]Guy's and St. Thomas's NHS Foundation Trust, London, UK. [22]Karolinska Institute, Stockholm, Sweden. [23]King's College London, London, UK. [24]Plymouth Hospitals NHS Trust, Plymouth, UK. [25]Norfolk and Norwich University Hospital NHS Foundation Trust, Norwich, UK. [26]Nottingham University Hospitals NHS Trust, Nottingham, UK. [27]University College London, London, UK. [28]Wythenshawe Hospital, Manchester, UK. [29]University Hospitals Coventry and Warwickshire NHS Trust, Coventry, UK. [30]Peterborough Hospitals NHS Trust, Peterborough City Hospital, Peterborough, UK. [31]Department of Surgery and Cancer, Imperial College, London, UK. [32]Queen's Medical Centre, University of Nottingham, Nottingham, UK. [33]Centre for Cancer Research and Cell Biology, Queen's University Belfast, Belfast, UK. [34]Tayside Cancer Centre, Ninewells Hospital and Medical School, Dundee, UK. [35]Portsmouth Hospitals NHS Trust, Portsmouth, UK.

## Methods

### Patient cohorts and study design

A multicenter consortium, Oesophageal Cancer Clinical and Molecular Stratification (OCCAMS), prospectively recruited a cohort of 2,115 patients with EAC on a curative pathway across 25 UK centers[18]. This included gastroesophageal junction cancers, but gastric cardia cancers were excluded. OCCAMS was registered and approved by relevant research ethics entities (East of England Ethics Committees, UKCRNID-8880, REC 07/H0305/52 and 10/H0305/1). Study sites were trained to include details of a prior history of BE and any evidence of BE on diagnosis. In addition, the endoscopic, surgical and histopathological reports were manually reviewed to classify each case phenotypically according to the presence or absence of macroscopic or microscopic evidence of BE metaplasia adjacent to the tumor. For precancerous BE samples used as a comparison, we used cases from the BEST2 and Barrett's Biomarker studies (East of England Ethics Committees 10/H0308/71, LREC01/149 and registered in the Integrated Research Application System (IRAS) (ID 57563, 15949)).

All participants in OCCAMS, BEST2 and Barrett's Biomarker studies provided individual informed consent, and data were pseudonymized.

### Selection of cases

The inclusion criteria for the OCCAMS study select patients with a confirmed diagnosis of adenocarcinoma of the esophagus and gastroesophageal junction who were fit enough for treatment on a curative pathway, which was generally neoadjuvant chemotherapy and surgery. For these cases, we aimed to collect pretreatment samples for sequencing, but, where this was not possible, a sample was taken from the surgical resection specimen. For patients with early disease, treatment comprised endoscopic therapy (endoscopic mucosal resection with or without radiofrequency ablation). A small number of patients with advanced-stage disease were included who were initially deemed to be curative but in whom the full staging showed more advanced disease not suitable for a curative pathway.

Comprehensive clinical research guidelines were developed in the Fitzgerald research group and followed by trained clinical and research staff at all OCCAMS study centers. At each study center, eligible patients with EAC were identified and approached regarding participation in the OCCAMS study and their desire to join as participants. Alternatively, OCCAMS research staff in the clinic asked the patient's permission to be contacted with more information via mail and a follow-up phone call. Patients had the opportunity to ask questions and think about their involvement by talking to their general practitioner, for example, and could return the signed form later. Consent was obtained from patients to contact their general practitioner to inform them of their participation in OCCAMS and to obtain relevant medical information from the cancer registries and other National Health Service (NHS) data controllers.

Patients with pathologically assessed tumors and diagnosed with adenocarcinoma of the esophagus between 2002 and 2022 were included. Patients with tumor histology other than EAC were excluded, and a majority (n = 233, 67%) were esophageal squamous cell carcinoma (ESCC) cases. Furthermore, patients with 'open & shut' surgery with more advanced disease than expected were also excluded. This was because a tumor sample was generally not collected for these patients; therefore, tumor phenotype ascertainment would have not been possible. In addition, few data were recorded on the baseline questionnaire forms for such patients. A small number of cases (n = 22, <1%) were missing age or gender, and these were excluded.

### Pathology review

A strict expert pathology review was performed for all cases. At least two pathologists reviewed each EAC case: one pathologist from the referring study center and another pathologist from the OCCAMS central study center at Cambridge University Hospitals who had more than 20 years of experience in upper gastrointestinal cancer. Tumors were

staged based on International Union Against Cancer/American Joint Committee on Cancer (UICC/AJCC) TNM guidelines (7th edition)[31]. The T, N and M stages were assigned using the available information in the patient's medical records, including clinical chart notes, endoscopic ultrasound, positron emission tomography, endoscopic mucosal resection (EMR) and histopathological reports after surgical resection. We used the most advanced stage prior to or at the time of surgery for patients who received neoadjuvant therapy.

The presence of BE adjacent to EAC for OCCAMS cases was based on one of two criteria—endoscopic (macroscopic) visual changes observed at prestaging evaluation with pathology review showing intestinal metaplasia at the time of surgical resection, which was assessed by expert gastrointestinal pathologists of the recruiting OCCAMS sites. Intestinal metaplasia was also identified in cases without macroscopic evidence of BE upon expert review of the pathology specimen. All pathologists followed a specific synoptic report proposed by the College of American Pathologists. Additionally, pathologists followed the OCCAMS study protocol, which required thorough assessment for BE in the proximal and distal resection margins and tumor. Tumor sampling was done for all borders of the resected tumor and the tumor bed to minimize sampling error. The number of biopsy specimens varied based on tumor size.

### Clinical data collection

Trained research staff collected baseline characteristics using chart review or during structured face-to-face interviews using a uniform case report form (CRF) across the 25 sites in the OCCAMS Consortium. All covariates used in the analysis originated from the study CRF. Patient baseline characteristics were collected on demographic, anthropometric and environmental exposures. Weight and height were measured objectively at the baseline visit or from the next closest record to the baseline. Gender was determined based on self-reported questionnaire data. Overall survival time (in years) was calculated from the date of diagnosis to the date of death or the date the patient was last seen in the clinic. Vital status was ascertained from all-cause mortality. All patients who consented to participate provided the minimum reporting standard, which required demographic and clinical details.

Research staff transcribed and entered data captured on the OCCAMS CRFs into the study database. These data were anonymized and stored in a secure central database hosted on Cambridge University Hospitals NHS Foundation Trust servers. Several data management issues should be noted as the data collection process may introduce errors or biases. Errors during the baseline interview (for example, failure to ask questions or record a response) or lack of information in the case notes/electronic records may contribute to missing data. As many patients are of advanced age, recall bias may also introduce discrepancies (for example, answers to history of heavy drinking or smoking).

### Data preparation and variable construction

**Processing of baseline clinical and epidemiological data.** The raw and fully anonymized baseline data for OCCAMS (R data file format, .Rdata) and BEST2 (comma-separated values file format, .csv) were exported to a university-furnished computing device in June 2022. The files containing the data for OCCAMS and BEST2 were collated, processed and screened for completeness, accuracy and consistency. Data were cleaned, removing or correcting any inconsistencies, inaccuracies or implausible values. All pragmatic strategies to minimize missing data were implemented. The cleaned dataset was then carefully checked against the raw data to ensure quality data preprocessing. The datasets and data-cleaning code were saved as plain-text files and tracked and managed using version control software (Git/Subversion). All data processing was carried out using R version 4.2.3 on macOS Ventura 13.3.1.

The following common methodology was used to clean data for both OCCAMS and BEST2 studies. Due to the inclusion criteria, age at

diagnosis and gender were complete. Ethnicity was recoded into 'white' or 'other' as there were too few observations in other ethnicity codes, which is not unexpected for the BE/EAC patient population.

The age at diagnosis for EAC and BE cases and age at recruitment for reflux controls, as well as BMI at baseline, were categorized into groups. This was done to create a more meaningful comparison for these measures. Age at diagnosis was categorized into four groups: younger than 50 years, 50–59 years, 60–69 years and 70 years or older. BMI was calculated using the baseline weight and height (kg m$^{-2}$), and BMI categories were defined according to standard ranges of underweight (<18.5), normal weight (18.5–24.9), overweight (25–29.0) and obese (≥30). Underweight cases were included among normal weight owing to very small frequencies in the cohort (<2.5%). The continuous distribution and the grouped frequencies were used in descriptive analyses, and only the categorical variables for age and BMI were included in regression models.

Cigarette smoking was collapsed into a binary variable, with 'former' and 'current' recoded as 'ever' smoker and 'never' remaining as defined. Additionally, if the average number of cigarettes per day was recorded as zero, smoking status was set to 'never'; if it was a non-zero value, then smoking was set to 'ever'. The number of pack-years was calculated by multiplying the number of packs of cigarettes smoked per day by the number of years of smoking.

The self-reported responses for medication use frequency of aspirin, non-steroidal anti-inflammatory drugs (NSAIDs), proton pump inhibitors (PPIs), H2-receptor antagonists (H2RAs) and over-the-counter (OTC) acid suppressants included 'Never', 'No', 'Past Use', 'Occasional Use' and 'Current Use'. However, responses such as 'Past Use' and 'Occasional Use' are open-ended, so, to mitigate this issue, responses were recoded to binary 'Ever Use' and 'Never Use'. The duration of medication use was recorded in years, months, weeks and days. The total duration of use (in years) was calculated for each medication type by summing the individual measures. Additionally, if the frequency was set to 'Never' and a non-zero total duration of use was reported, the total was set to null. Conversely, if a non-zero duration was recorded, then the frequency of use was set to 'Ever Use'. Aspirin and NSAID frequency of use were combined into a single variable measuring use of either medication. Similarly, a single variable for the use of any acid suppressant medication was derived using the frequency of use of PPIs, H2RAs and OTC acid suppressants.

Alcohol intake was recorded as the number of units of beer, wine and spirits consumed per week. These individual measures were summed into a single continuous variable for the total number of alcoholic drink units consumed per week. Heavy drinking status was self-reported by patients in both studies.

Frequency of reflux symptoms was reported as 'Never', 'Sometimes', 'Often', 'Daily' and 'Unknown/sporadic'. Duration of reflux symptoms was harmonized into an ordinal variable with four ranges (Never, 5 years, 5–10 years, >10 years and Unknown/sporadic). A single variable was created that combined all measures related to reflux symptoms and acid suppressant medication use. This variable is referend to as the 'derived heartburn symptoms status' variable (Supplementary Fig. 3). In addition, a single variable for use of any acid suppressant medications was derived based only on the acid suppressant medication use variables (patient on acid suppressant and use or duration of PPIs, OTC acid suppressant medications or H2RAs).

For variables that contained responses with undefined free text or numeric ranges instead of a single value, either the response was set to missing or the mid-range was calculated. For example, a free text input of 'undistilled only' for total alcohol unit intake was set to missing, and a response of '3–5' cigarettes per day was recalculated to '4' per day. Continuous variables where a negative numeric value was recorded were recoded to missing as per CRF instructions.

The UK regions for OCCAMS and BEST2 study centers were determined based on their locations and classified using the International Territorial Level 1 (Office of National Statistics). Finally, for EAC cases only, combined TNM staging was created according to UICC/AJCC 7th edition guidelines[31].

**Variable selection.** After baseline data cleaning and screening in the OCCAMS cohort and as informed by the results of the literature review, a total of 32 variables across five domains were deemed relevant and included (Supplementary Table 2). To select variables for inferential analysis, a purposeful selection process was followed:

1. Unconditional logistic regression was used to obtain univariable odds ratios and 95% confidence intervals for the association of each variable with BE-negative EAC compared to BE-positive EAC cases.
2. Variables with $P < 0.25$ and missing data <60% overall were preselected and included in a multivariable logistic regression model with BE-negative EAC as the outcome compared to BE-positive EAC.

Only variables with $P < 0.05$ or those deemed to have epidemiological or clinical importance were selected in the final stage. Directed acyclic graphs were also used to determine which variables should be included as potential confounders.

**WGS.** WGS with a matched germline reference was available for one EAC sample from each of 710 patients and for 388 samples across 256 patients with BE. WGS was performed on as many cases as possible, but, prior to inclusion, each sample had to pass a strict pathology consensus review on a fresh-frozen section and show cellularity greater than 70%. Methods for sample quality control, DNA extraction and WGS were as previously described[12,19], and cohort details, single-nucleotide variant (SNV), copy number and structural variation analysis are detailed as follows.

**BE cohort.** The BE cohort includes 388 samples with 205 sequenced from single BE samples and 79 pooled sequencing from 183 multilevel samples. There are sequencing data from 284 endoscopies of 256 patients in total. For those pooled for sequencing, we took the highest pathology as the grade annotation. This cohort was selected to capture early genomic events preceding invasive EAC. Specifically, there are 51% (145/284) non-dysplastic Barrett's esophagus (NDBE) and 49% (139/284) dysplasia, including 24% (68/284) low-grade dysplasia (LGD) and 25% (71/284) HGD, including very few T1a samples. Patients with NDBE as their highest pathology were followed for a median of 3.1 years (IQR: 2.2–4.4) until their last surveillance endoscopy. Most dysplastic lesions were treated at diagnosis, whereas a few historical untreated LGD cases were followed for a median of 1.3 years (IQR: 0.8–2.7). These precancerous patients were from the BEST2 and Barrett's Biomarker studies (East of England Ethics Committees 10/H0308/71, LREC01/149 and registered in the IRAS (ID 57563, 15949)).

**BE and EAC sample processing.** Strict pathology consensus review was observed for these BE and EAC samples, with a minimum 70% cellularity requirement before inclusion. All tissue samples were snap frozen. For the OCCAMS study, peripheral blood was used as the germline reference, and, in cases where this was not possible, a sample of normal squamous epithelium located at least 5 cm away from the lesion or normal duodenal tissue was used instead, according to standard practice.

Methods for sample quality control, DNA extraction and WGS were as previously described[12,14,19]. In brief, the 710 EAC and 205 BE samples sequenced by Illumina, the Cancer Research UK Cambridge Institute and the Wellcome Sanger Institute underwent WGS to a target depth of 50×. Matched germline samples were sequenced to a target depth of 30×. Reads were then aligned with BWA-MEM to the 1000 Genomes

Project version of the GRCh37 human reference genome[32]. Each of the 79 BE sample pools and matched germline samples sequenced by Genomics England were processed in two aliquots to combined target depths of 150× and 75×, respectively, and reads were aligned to GRCh38. Sequencing quality checks were conducted using the FastQC package (https://www.bioinformatics.babraham.ac.uk/projects/fastqc/) and polymerase chain reaction (PCR), and optical duplicates were flagged using Picard MarkDuplicates (https://broadinstitute.github.io/picard/) after alignment.

**Single-nucleotide and copy number variant calling.** For the 710 EAC and 205 BE samples, somatic variants were called using Strelka version 2.0.15 (ref. 33) and annotated using Variant Effect Predictor (VEP) version 78 (ref. 34). Mutation burden was derived from each sample's VEP files by summing the number of SNVs and indels across the genome. For the additional 79 BE samples sequenced by Genomics England, Strelka version 2.9.4 and VEP version 91 were used, and mutation burden was calculated by taking the average across the aliquot VEP files for each sample. LiftOver was used to convert mutation loci between versions of the human reference genome. Mutations per megabase was calculated using the length of the reference genome (3,137,454,505 bp). GISTIC2.0 was used to detect recurrently deleted or amplified regions of the genome using raw copy number values obtained from ASCAT version 2.1 (refs. 35,36) for the 710 EAC and 205 BE samples and from Canvas version 1.38.0.1554 for the 79 BE samples.

**Selection and calling of driver genes.** Previously reported driver genes in EAC were derived from genes listed in Frankell et al.[19], and genomic regions were identified using Ensembl BioMart[37]. These gene regions were then used to extract alterations from the outputs of VEP and GISTIC2.0. Driver mutation status was determined based on the alteration type (for example, missense, nonsense or frameshift) using Strelka and VEP. One or more affected copies were deemed as a mutation.

To identify driver genes associated with BE, we predicted SNV mutations using observed/expected mutation ratios calculated by dNdScv. Copy number driver genes were identified by overlapping genes located within peak regions detected by GISTIC2.0 with those in the COSMIC consensus and previously identified EAC driver genes.

**Mutational signatures.** Mutational signatures discovery within the cohort was carried out using SigProfilerExtractor[38] on 997 samples as previously described[12]. The optimal signature configuration was determined by selecting from a range of signature combinations (from five to 17) based on the highest stability and lowest Frobenius reconstruction error for a signature combination. The optimal configuration was composed of 14 signatures, and its validity was confirmed by independent analysis using Bayesian methodology from sigminer[39]. Subsequently, deconstructSigs[40] was employed to deduce the mutational contributions of these processes to each sample across the entire cohort presented here.

**Copy number, whole-genome duplication and aneuploidy.** An amplification is defined as a ploidy-adjusted copy number greater than 2, and a deletion is defined when the copy number value is 0. The percentage of aberrant genome is calculated as the proportion of the genome, excluding sex chromosomes, where the rounded copy number does not equal the rounded ploidy. The fraction of loss of heterozygosity is defined as the percentage of the genome where the minor allele frequency is less than 0.5, relative to the entire genome excluding sex chromosomes. Raw copy number values from ASCAT and the PCAWG-11 consensus purity pipeline (https://github.com/PCAWG-11) were used to determine samples with whole-genome duplication based on tumor ploidy and the extent of loss of heterozygosity[41]. Per-sample ploidy and purity were also inferred using this method.

**Identification and classification of amplicon and chromothripsis events.** Copy number segments were called using CNVkit version 0.9.8, and regions of amplifications of size 50 kb and copy number >4.5 were used as input for the identification of amplified regions and reconstructed using AmpliconArchitect[42,43].

The classification of amplicons into ecDNA and BFB events was done using AmpliconClassifier (https://github.com/AmpliconSuite/AmpliconClassifier). Chromothripsis events were performed using ShatterSeek (https://github.com/parklab/ShatterSeek). High-confidence chromothripsis events were defined as ≥6 interleaved intrachromosomal structural variants, ≥7 contiguous segments oscillating between two copy number states and fragment joins testing positive, plus either chromosomal enrichment or exponential breakpoint distribution. Low confidences were defined as the same structural variant criteria but 4–6 segments oscillating[44].

**WES.** WES was conducted and analyzed in 380 samples from 87 patients with multiregional chemotherapy-naive tumor samples (Extended Data Fig. 4a). Samples were obtained from the macrodissection of FFPE esophagectomy or esophagastrectomy specimens (approximately 0.5 × 0.5 cm). The tumor samples were required to be spatially distinct, so the number of tumor samples depended on the size, quality and spatial arrangement on the specimen. Only specimens that yielded at least two tumor samples were included, and the number of tumor samples ranged from two to six, with a mean of 4.3. The specimens covered all clinical stages, with a skew toward later stage, with 61% being stages III and IV, in keeping with a generally late stage of presentation. FASTQ files were aligned to GRCh37 using BWA-MEM, with duplicates marked by Picard version 2.9.5. Variant calling was performed using GATK Mutect2 version 4.1.7.0, using multisampling and FFPE settings.

**Evolutionary trajectory and clonality analysis of driver genes.** Somatic mutation clustering was performed using PyClone[45] (version 0.13.1) to identify subclonal populations based on SNV data. Clonal phylogenies were reconstructed using ClonEvol[46] (version 0.99.11) on the PyClone-derived clusters, excluding clusters with fewer than five mutations. Indels and copy number driver events were added to the trees post hoc. REVOLVER[47] (version 1.0.0) was used to determine trajectories of repeated evolution from multiple regions. The number of clonal and subclonal driver and passenger mutations was assessed using dNdScv[48] based on the output of PyClone.

To evaluate intratumour heterogeneity (ITH) and clonal architecture across EAC phenotypes, we analyzed and compared the following metrics using the Wilcoxon rank-sum test.

- Mutational ITH: ratio of subclonal SNVs to total SNVs
- Total driver mutations: non-silent SNVs/indels in previously reported EAC driver genes
- Total mutations: non-silent SNVs/indels
- Recent subclonal expansion score: the largest terminal node cancer cell fraction (CCF) across all samples in a case—calculated for every potentially valid tree, with the minimum score used in analysis
- Clones with driver mutations: number of clones harboring ≥1 driver gene alterations
- Truncal mutations: silent and non-silent SNVs present in the founding (truncal) clone
- Truncal driver mutations: non-silent SNVs/indels in driver genes located in the truncal clone

**Spatial transcriptomics.** Spatial transcriptomics landscape of BE and EAC was measured in seven samples (five EAC and two BE; Extended Data Fig. 5a) from six patients with EAC. Samples were fresh-frozen chemotherapy-naive tumor biopsies with well-preserved tissue morphology. Spatial transcriptomics profiling was performed

using the 10x Genomics Visium HD WT platform measuring approximately 18,000 genes, which achieves 2-µm resolution, enabling near-subcellular spatial granularity. Frozen tissue was selected over FFPE for Visium analysis because well-preserved fresh-frozen samples generally yield higher-quality RNA and improved transcriptomic signal[49,50]. We also confirmed that tissue morphology was well preserved in the frozen sections used for this study.

**Cohort design and spatial transcriptomics experiment.** To investigate the spatial transcriptomics landscape of BE and EAC, treatment-naive patient samples were selected from the OCCAMS database in which fresh-frozen biopsies had been collected at endoscopy and stored at −80 °C. The primary selection criterion was well-preserved tissue morphology, essential for accurately correlating transcriptomic profiles with histological features.

Given that fresh-frozen tissue may exhibit variable morphological preservation compared to FFPE specimens, all biopsies underwent blinded, standardized pathological evaluation. Hematoxylin and eosin (H&E)-stained sections were reviewed by a gastrointestinal pathologist, and overall morphology was classified as good, medium or poor based on cellularity, the presence of well-differentiated tumor epithelium in EAC cases or clearly identifiable epithelial structures in BE cases and the absence of extensive necrosis, hemorrhage or other forms of tissue degradation.

After this review, seven samples were analyzed for spatial transcriptomics (Extended Data Fig. 4a). Profiling was performed using the 10x Genomics Visium HD platform, which achieves 2-µm resolution for near-subcellular spatial granularity. All histological processing was performed by the Histology Core at the Cancer Research UK Cambridge Institute. Fresh-frozen esophageal biopsy samples were cryosectioned to 10-µm thickness. Libraries were prepared using the Visium HD Gene Expression Slide and Reagent Kit (4 rxns, PN-1000675) (10x Genomics) and sequenced. Sequencing was performed on an Illumina NovaSeq X system at a depth of approximately 300 million reads per sample, sufficient for 2-µm resolution. Raw sequencing data were processed using Space Ranger (version 3.1.3) (10x Genomics) with alignment to the GRCh38 human reference genome. Spatial gene expression matrices were generated and visualized using Loupe Browser (version 8.1.2). Each Visium HD spatial transcriptomics map was co-registered with the corresponding H&E-stained histology image using ImageJ software.

**Unsupervised clustering.** To identify spatial transcriptional domains within the tissue, unsupervised clustering was performed using the Leiden algorithm[51] and overlaid on the tissue image. To assess whether the transcriptionally defined clusters corresponded to histologically distinct regions, H&E-stained slides annotated with cluster assignments were reviewed by a board-certified pathologist. Cluster composition and spatial distribution were compared between BE and EAC samples to examine conserved and stage-specific transcriptional patterns.

Differential gene expression analysis was conducted using the Scanpy package (version 1.11.1) and applied to Leiden clusters[52]. The Wilcoxon rank-sum test, as implemented in Scanpy, was applied for pairwise comparison. Genes were ranked according to test statistics, and those with an adjusted P value lower than 0.05 were considered significant markers. Multiple testing correction was performed using the Benjamini–Hochberg FDR method[53].

*Profiling with BE and EAC marker genes.* Cell type annotation was performed using established marker genes. Literature-derived gene panels were assembled to identify the major cell populations present in EAC, columnar epithelium, squamous epithelium, stromal and immune populations (Extended Data Table 5). Tissue type annotations were based on the read counts of marker genes, with squamous or stromal tissues assigned according to the dominant expression.

For EAC and BE classification, regions expressing EAC-specific genes were designated as 'EAC regions', whereas regions expressing only BE-specific genes were labeled as 'BE regions'. Bins were colored based on tissue type, with more dense regions reflecting a greater number of bins expressing relevant genes and lower density indicating less-abundant expression.

**IHC analysis of BE protein markers.** TFF3 and REG4 were assessed in FFPE samples (biopsies, EMRs and surgical resections) from 18 patients with EAC (eight BE-negative and 10 BE-positive). Sections (4 µm) were stained using a BOND RX automated stainer (Leica) and scanned at ×40 magnification (Leica, Aperio AT2).

TFF3 and REG4 biomarkers for the histopathological hallmark of intestinal metaplasia[20,54] were immunohistochemically assessed. FFPE samples from endoscopic biopsies, EMRs and surgery were obtained from a subset of patients from the Cambridge cohort who were enrolled onto the OCCAMS study. A total of 31 sections from eight patients with BE-negative EAC (one moderate, three moderate to poorly differentiated and four poorly differentiated) and 72 sections from 10 patients with BE-positive EAC (one well differentiated, four moderate and five moderate to poorly differentiated) were stained, with matched age and gender. Pathologist evaluation was used to determine specimens of interest in surgical samples and adjacent and/or longitudinal preneoplastic lesions including BE using H&E staining. All samples were sectioned at 4-µm thickness, and IHC staining was carried out using BOND RX (Leica automated stainer). For TFF3 (Invitrogen), the heat-induced epitope retrieval (HIER) used was 1, with 10 minutes in HIER, and the antibody was used at a dilution of 0.26 µg ml⁻¹. For REG4 (Abcam, ab255820), the HIER used was 2, with 10 minutes in HIER, and the antibody was used at a dilution of 0.26 µg ml⁻¹. All images were acquired using an Aperio AT2 slide scanner (Leica) at ×40 magnification. Pathologist review was performed in all staining interpretations.

### Statistical analysis
**Logistic regression.** Unconditional logistic regression was used to obtain univariable odds ratios and 95% confidence intervals for the association of each variable with BE-negative EAC compared to BE-positive EAC cases. Variables with $P < 0.25$ and missing data <60% overall were preselected and included in a multivariable logistic regression model with BE-negative EAC as the outcome compared to BE-positive EAC. For each comparison set, crude and adjusted odds ratio and 95% confidence interval were obtained. We performed four separate adjusted analyses per comparison: (1) minimally adjusted for age and gender only, (2) fully adjusted for all covariates, (3) fully adjusted model eliminating heartburn as a covariate and (4) tumor stage adjusted model. We then developed a logistic regression model for 435 patients, integrating the most relevant genomic and clinical data to understand whether the two phenotypes could be differentiated by integrating both datasets.

The statistical independence of the outcomes was assumed based on the absence of repeated events and the binomial distribution of the residual variation. It is rare for this assumption of logistic regression to be violated. To ensure that the assumption of multiplicativity was satisfied, effect measure modification was assessed between BMI and heartburn and between aspirin/NSAID use and heartburn. A priori, it was known that BMI may modulate heartburn. The latter interaction was tested because the heartburn variable was partly derived from PPI use, and NSAIDs may modify the effects of PPIs in relation to EAC[55].

As heartburn may be on the causal pathway, if its elimination as a covariate changed the log odds ratio by more than 10%, then it could be considered a confounder. Missing data were coded as indicator variable.

Three sensitivity analyses were performed to assess the robustness of the estimates obtained using the fully adjusted model for each comparison set. The first sensitivity analysis involved excluding any

observations with missing data for the variables in the fully adjusted model, adopting a complete case approach. The second sensitivity analysis used estimates derived from multiple imputation data (detailed below). Lastly, a sensitivity analysis was conducted by excluding EAC cases with a history of undergoing BE surveillance.

No statistical method was used to predetermine sample size. However, this study represents, to our knowledge, the largest clinical and genomic cohort to date, with the next largest cohorts having 54% fewer patients with clinical[11] and 31% and 49% fewer patients with EAC and BE genomically sequenced, respectively[14,19]. We did not exclude any data from the analyses. The experiments were not randomized. The pathologist investigators were blinded to the risk factor data (heartburn, smoking and BMI). Other investigators were not blinded to the independent variables and phenotype outcome.

**Missing data.** Missing data for baseline characteristics were calculated as a percentage of the total number of cases. The percentage of the recorded values is reported as a fraction of complete cases. For variables dependent on the response to other variables, the missing percentage was calculated as a fraction of cases where the first response variable was available. For example, the proportion of missing data for the duration of cigarette smoking was based on the total number of patients who self-reported current or former cigarette smoking.

Multiple imputation was performed on the datasets corresponding to each comparison group to assess how missing data might bias the observed associations. Age and gender were complete and, therefore, not included in multiple imputation. BMI group, cigarette smoking, aspirin/NSAID use, heartburn symptoms and TNM (EAC only) were imputed using multiple imputation by chained equations with the appropriate method selected based on the variable type[56].

The missing data were assumed to be missing completely at random, meaning that the probability of a value being missing is not related to other data. This assumption was based on the similar distribution observed for recorded and imputed data. Furthermore, baseline data were collected by numerous research staff, and, based on our experience, we assumed that variations in the order of the CRF questions, completeness of each section and other factors may have impacted the quality and accuracy of the data collected. Therefore, we assumed that systematic exclusion of data was unlikely. The number of imputations ($m$) was set to the percent value of the variable with the highest amount of missing data in each dataset, which was aspirin/NSAID use, with approximately 50–60% missing data. The number of iterations ($n$) was set to 20, as typically recommended[57].

**Multiple correspondence analysis (MCA).** To delineate the genomic differences between BE-positive and BE-negative EAC, we employed MCA focusing on mutations in recognized EAC driver genes. Each mutation type was categorized distinctly to enable comprehensive analysis. This analysis was performed using the FactoMineR package in R, a robust tool for multivariate exploratory data analysis. Visualization of the MCA results was accomplished using the fviz_mca_ind function from the factoextra package.

**Non-parametric data, transformations and multiple hypothesis testing.** Statistical comparisons between groups were performed using either the Kruskal–Wallis test or the Mann–Whitney $U$-test, as indicated by the normality of the data distribution. When applicable, data were log transformed to ensure normality. The percentage of driver genes between groups was compared with $\chi^2$ testing. In cases where multiple comparisons were made, adjustment for FDR using the Benjamini–Hochberg procedure or Bonferroni correction were applied.

**Computing environment.** All analyses were performed using R version 4.2.3 on macOS Ventura 13.3.1 with packages 'rstatix', 'mice', 'survival', 'survminer' and 'coxme'.

## Reporting summary

Further information on research design is available in the Nature Portfolio Reporting Summary linked to this article.

## Data availability

The data that support the findings of this work are available as follows. WGS data for samples sequenced by Illumina or the CRUK Cambridge Institute are available at the European Genome-phenome Archive (EGA) under accession number EGAD00001015435. WGS data for samples sequenced by the Wellcome Sanger Institute are available at the EGA under accession number EGAD00001006083. WGS data for samples sequenced by Genomics England are available in the Genomics England National Genomic Research Library (https://www.genomicsengland.co.uk/research/data). Source data are provided with this paper.

## Code availability

All bioinformatic tools, versions and custom codes in this work are publicly deposited at https://github.com/fitzgerald-lab/EAC-Phenotypes.

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

## Acknowledgements

This work was supported by Cancer Research UK (CRUK) (A15874, A22720 and A22131) and the Medical Research Council (MR/W014122/1) to R.C.F., with additional support by the UK National Institute for Health Research (NIHR) Cambridge Biomedical Research Centre (NIHR203312). This work was also supported, in part, by the Bill and Melinda Gates Foundation Gates Cambridge Scholarship to S.A.Z. (OPP114), the Jack Kent Cooke Foundation Graduate Scholarship and the National Cancer Institute (NCI) Cancer Prevention Fellowship Program. L.W. was supported by a China Scholarship Council Postdoctoral Program fellowship. M.S. was supported by a UK Research and Innovation (UKRI) Future Leaders Fellowship (MR/Y034031/1). This work uses data provided by patients and collected by the NHS as part of their care and support. The views expressed are those of the authors and not necessarily those of the NIHR or the Department of Health and Social Care. The contributions of the National Institutes of Health (NIH) author were made as part of their official duties because NIH federal employees are in compliance with agency policy requirements, and their contributions are considered works of the US government. The findings and conclusions presented in this paper, however, are those of the authors and do not necessarily reflect the views of the NCI, the NIH or the US Department of Health and Human Services. We thank the Human Research Tissue Bank of Addenbrooke's Hospital and the NIHR Cambridge Biomedical Research Centre for assistance with biospecimen sourcing and processing, the CRUK Experimental Cancer Medicine Centre for data infrastructure support and E. Steed for administrative assistance.

## Author contributions

R.C.F. designed the study, supervised the work and obtained funding and is guarantor for the content. S.A.Z. performed primary analysis for epidemiology data, and L.W. performed primary analysis for genomic data. E.L.B., A.B., A.W.T.N., M.S., D.J., J.P. and A.U. performed additional primary data analyses. N.G., B.N., A.F., G.D. and E.L.O. generated and cleaned the data. A.M. and M.O. provided pathology phenotyping expertise and pathological expert review. A.F. and S.K. supervised genomic analyses, and H.G.C. supervised epidemiological analyses. S.A.Z., L.W. and R.C.F. wrote the manuscript. All authors critically reviewed the manuscript.

## Competing interests

S.A.Z., L.W., E.L.B., A.B., A.W.T.N., M.S., D.J., G.D., N.G., B.N., A.F., A.M., S.K. and H.G.C. declare no competing interests. R.C.F. and M.O. are named on patents for Cytosponge and related biomarker assays, including those licensed by the Medical Research Council to Medtronic (formerly Covidien), and they own shares in Cyted Health Ltd. (company no. 11478299). S.K. did the work when in academia but is now an employee of Cyted Health Ltd. A.M.F. is a co-inventor on a patent application to determine methods and systems for tumor monitoring (PCT/EP2022/077987). These technologies are not discussed in this paper.

## Additional information

**Extended data** is available for this paper at https://doi.org/10.1038/s41591-026-04331-8.

**Correspondence and requests for materials** should be addressed to Shahriar A. Zamani or Rebecca C. Fitzgerald.

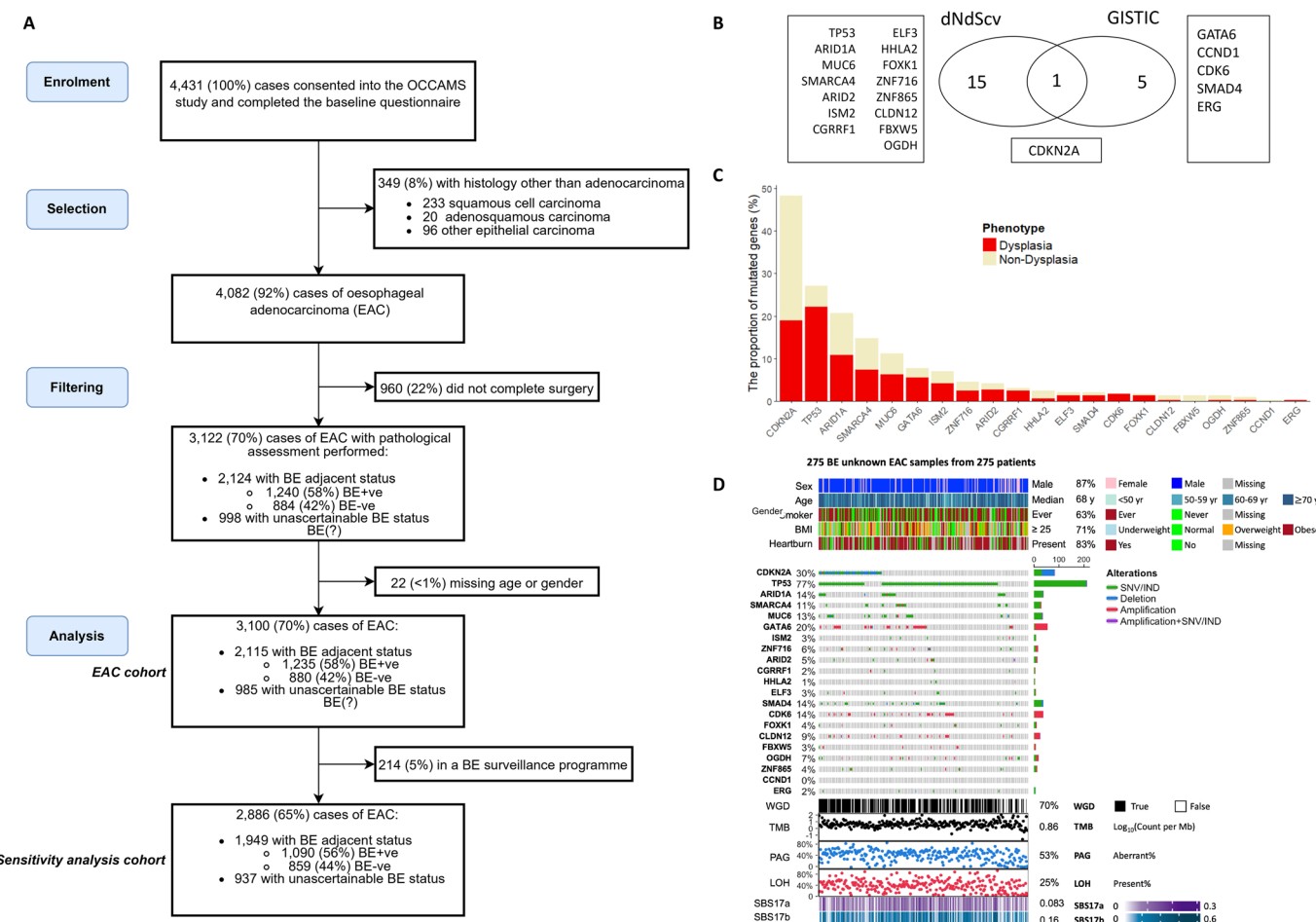

**Extended Data Fig. 1 | OCCAMS case selection and identification of Barrett's esophagus driver events, with mutational prevalence, epidemiological and genomic features. A**. Flow diagram for selecting esophageal adenocarcinoma cases. All cases originated from the OCCAMS database. Abbreviations: OCCAMS, Oesophageal Cancer Classification and Molecular Stratification; BE, Barrett's esophagus. **B**. Identification of driver genes in Barrett's esophagus (BE) cohort. We used dNdScv and GISTIC 2.0 to identify potential mutation and copy number driver genes in the BE cohort. These are variations in genes that promote clonal expansion and are positively selected for the growth of the lesion. dNdScv pinpointed 16 genes as drivers, including TP53, CDKN2A, ARID1, et al. GISTIC 2.0 identified an additional five genes, including GATA6, CCND1, CDK6, etc. **C**. The proportions of mutated BE driver genes in dysplastic and non-dysplastic samples. **D**. The epidemiological and genomic features of BE unknown esophageal adenocarcinoma (EAC). As a supplementary of Fig. 2, this panel shows the high-risk epidemiological factors associated with BE, including gender, age, smoking status, BMI, and heartburn history, genetic landscape of BE driver genes identified in our cohorts, and the prevalence of whole Genome Doubling (WGD), Tumour Mutational Burden (TMB), Percentage of Aberrant Genome (PAG), Loss of Heterozygosity (LOH), Signature of Base Substitution 17a (SBS17a), and Signature of Base Substitution 17b (SBS17b).

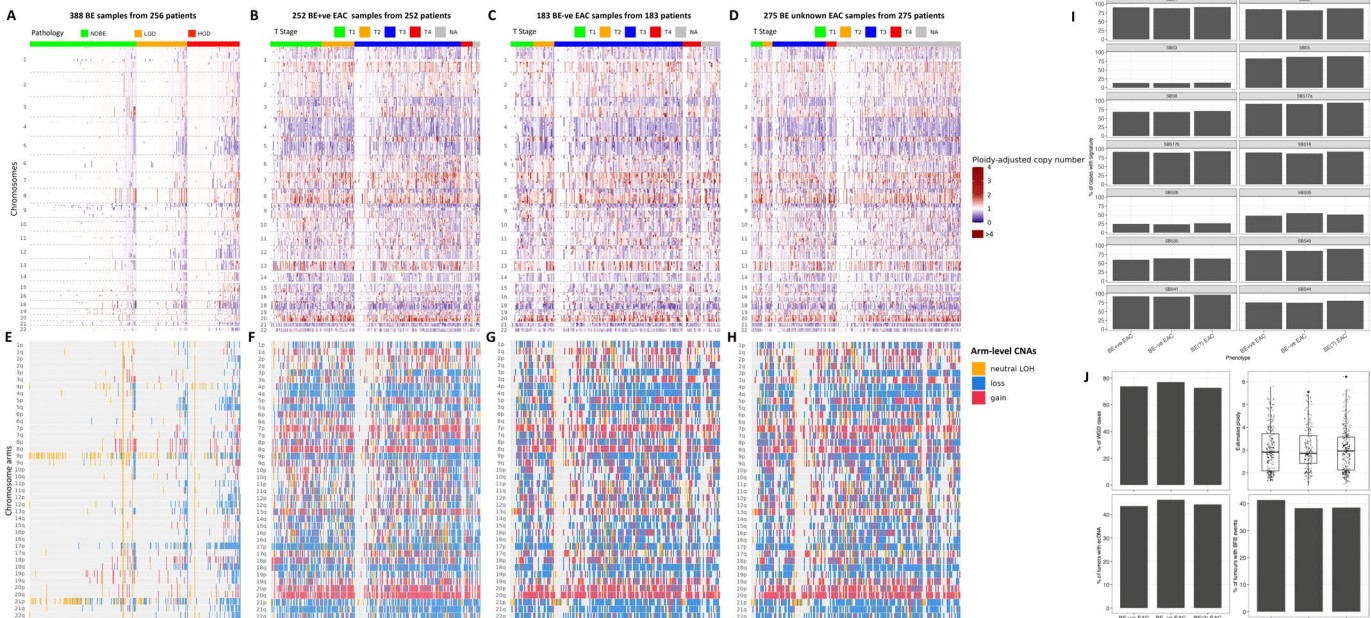

**Extended Data Fig. 2 | Genome-wide and arm-level copy number landscapes stratified by Barrett's esophagus (BE) pathology and EAC T stage, with mutational signature prevalence and large-scale genomic events across esophageal adenocarcinoma (EAC) phenotypes.** Genome-wide segmented copy number profiles (**A-D**) and arm-level copy number profiles (**E-H**) clustered by BE pathology and EAC T stage. **I**. The proportion of EAC cases harboring mutational signatures according to phenotype. Samples: total 710 including 252 BE positive (BE+ve) EAC, 183 BE negative (BE-ve) EAC and 275 BE unknown (BE (?)) EAC. The mutational signatures were uniformly distributed across both phenotypes. **J**. The distribution of large-scale and catastrophic events across esophageal cancer (EAC) cases, including the proportions of samples with whole-genome duplication (WGD), the distribution of estimated ploidy using the PCAWG-11 pipeline, the proportion of samples harboring extrachromosomal DNA (ecDNA), the proportion of samples with Breakage-Fusion-Bridges (BFB) events. Samples: total 710 including 252 BE+ve EAC, 183 BE-ve EAC and 275 BE(?) EAC. The boxplot represents the median (central horizontal line), quartiles 1 and 3 (bottom and top horizontal lines), and the minima/maxima (whiskers).

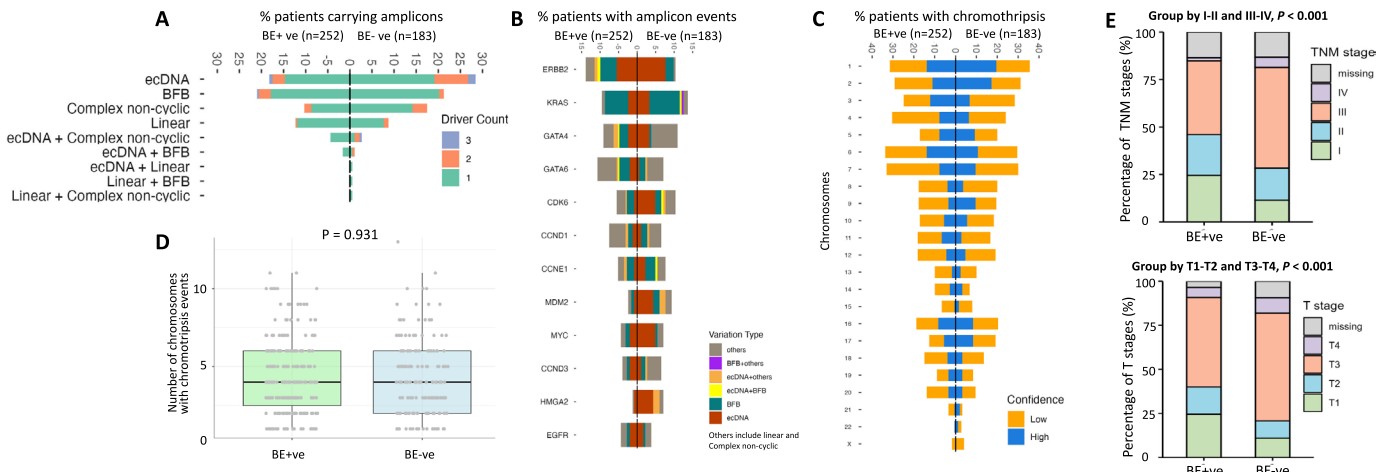

**Extended Data Fig. 3 | Catastrophic amplification and chromothripsis landscapes in esophageal cancer (EAC), with amplicon driver gene associations and stage distributions across Barretts esophagus (BE) positive (BE+ve) and BE negative (BE−ve) phenotypes. A.** Catastrophic events of amplicons and chromothripsis. Proportion of patients with amplicons involving extrachromosomal DNA (ecDNA), breakage–fusion–bridge (BFB), complex non-cyclic, linear, or combinations thereof, coloured by the number of driver genes. No significant difference was observed between BE+ve and BE−ve EAC. Two-sided Chi-square analysis with Bonferroni corrections for multiple comparisons was applied. **B.** Proportion of patients with each amplicon type per gene. MDM2 and HMGA2 showed borderline higher prevalence in BE−ve EAC (Adjusted P = 0.037 and 0.035) Two-sided Chi-square analysis with Bonferroni corrections for multiple comparisons was applied. MDM2 is a known TP53 inhibitor, and HMGA2 promoted MDM2-mediated p53 ubiquitination and degradation. However, the overall prevalence is low ( < 10%), and the higher frequency in BE−ve EAC is likely

related to its more aggressive nature rather than indicating a distinct pathway. **C.** Proportion of patients with chromothripsis. High confidence: ≥6 interleaved intrachromosomal SVs, ≥7 contiguous segments oscillating between 2 CN states, fragment joins test positive, plus either chromosomal enrichment or exponential breakpoint distribution. Low confidence: same SV criteria but 4−6 segments oscillating. No significant difference was observed between phenotypes. Two-sided Chi-square analysis with Bonferroni corrections for multiple comparisons was applied. **D.** Number of chromosomes with chromothripsis per patient, showing no significant difference between BE+ve (n = 252) and BE−ve EAC (n = 183). Box plots are presented as median and quartile range. Two-sided Wilcoxon test was performed without adjustment for multiple comparisons. **E.** The TNM and T stages across different phenotypes, showing that the Barrett's oesophagus (BE) negative phenotype is significantly more advanced than the BE +ve phenotype (Stage I-II vs. III-IV, P < 0.001; Stage T1-T2 vs. T3-T4: P < 0.001).

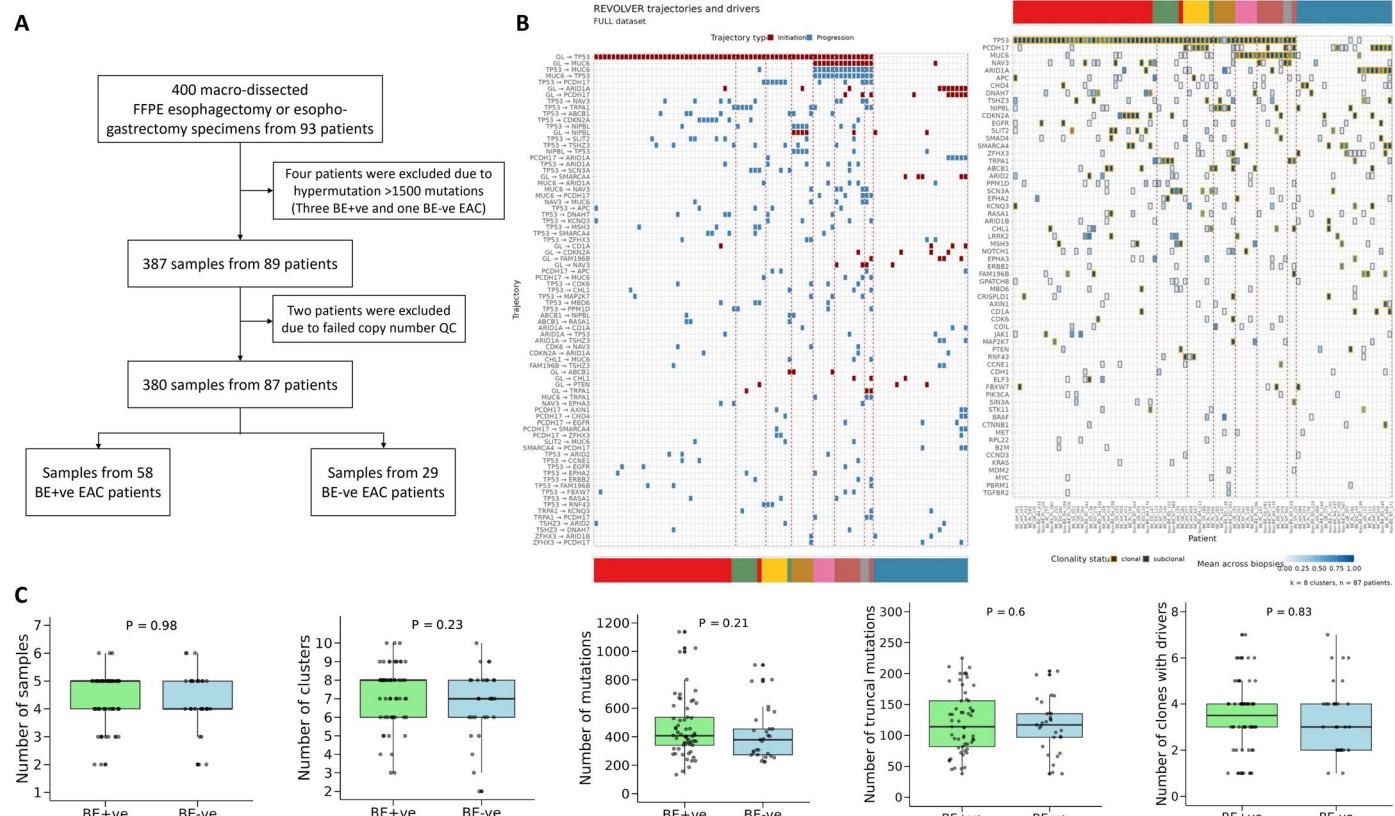

**Extended Data Fig. 4 | Whole-exome sequencing processing workflow and inferred evolutionary trajectories in multi-regional esophageal cancer (EAC), with comparative tumour heterogeneity metrics across Barretts oesophagus (BE) positive (BE+ve) and BE negative (BE−ve) phenotypes. A**. Workflow for data pre-processing of the whole-exome sequencing cohort. **B**. The full list of evolutionary trajectories and driver genes inferred from the multi-regional whole exome sequencing (WES) cohort. **C**. Supplementary metrics measuring tumour heterogeneity (BE+ve EAC: n = 58 and BE−ve EAC: n = 29). There is no significant difference between the two phenotypes of EAC. Box plots are presented as median and quartile range. Two-sided Wilcoxon test was performed without adjustment for multiple comparisons.

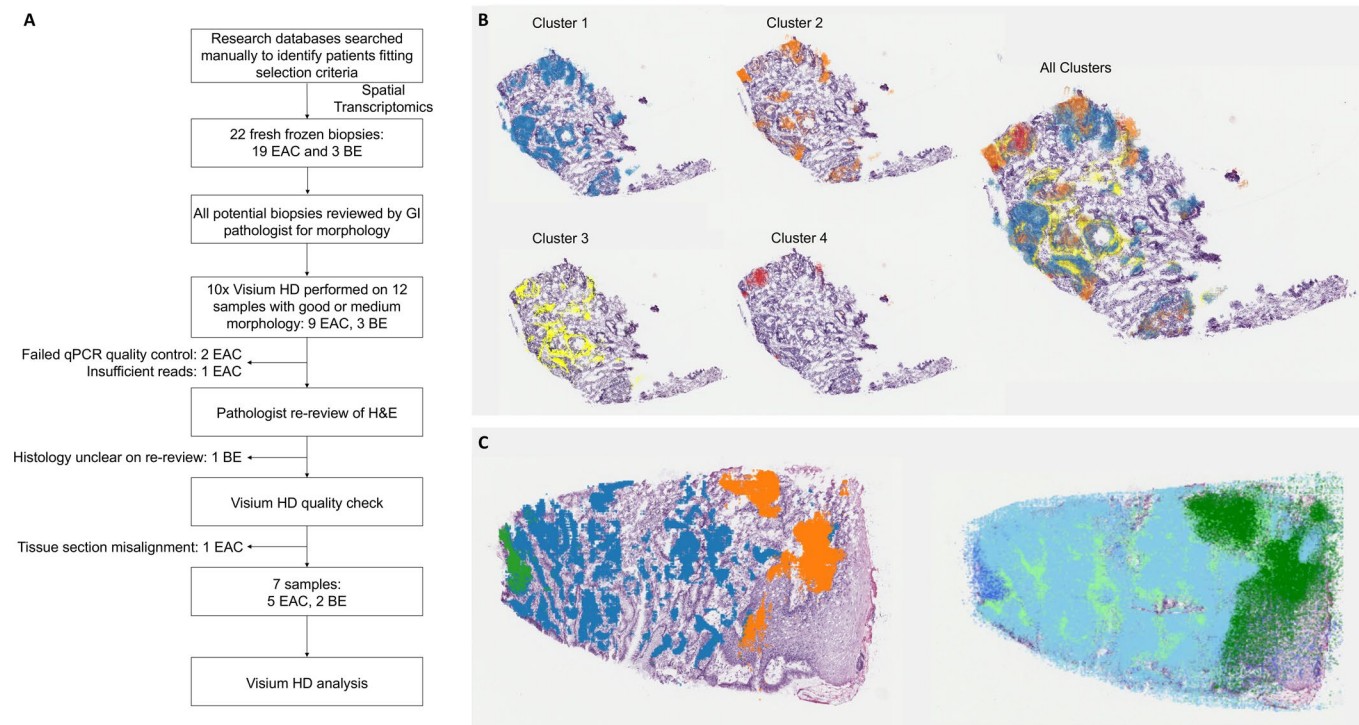

**Extended Data Fig. 5 | Spatial transcriptomics patient selection and spatial organization of transcriptional programs in esophageal cancer (EAC) and Barretts esophagus (BE), comparing unsupervised clustering with morphology-informed tissue compartment annotation. A**. Patient selection workflow for spatial transcriptomics analysis. **B**. Spatial localization of transcriptional clusters in an EAC sample. Individual panels show the spatial distribution of the four primary Leiden clusters overlaid on the H&E-stained tissue section. Cluster 1 (blue) represents the main proliferative tumor compartment, marked by secretory and cell cycle–associated genes including REG1A and TOP2A, notably BE lineage genes were also markers of this cluster in EAC samples with and without adjacent BE. Cluster 2 (orange) corresponds to cytokine-producing tumor regions enriched for CCL20, CXCL8, and DUOX2. Cluster 3 (yellow) localizes to fibroblast-rich stromal regions, expressing ECM-associated genes such as COL1A1 and SPARC. Cluster 4 (red) highlights spatially confined tumor-immune interface regions, characterized by CD74, CXCL5, and MMP3. Composite overlay of all clusters demonstrates the mosaic-like spatial organization of transcriptional programs within the tissue, cluster 3 is represented in yellow for better visualization. Distinct zones of proliferative, inflammatory, stromal, and immune-interacting activity emerge. This case is shown as a representative example illustrating how spatial clustering resolves modular transcriptional architecture within an EAC sample. **C**. Comparison between unsupervised transcriptomic clustering and morphology-informed tissue classification in BE. Unsupervised transcriptomic clustering of the same tissue section identifies four distinct domains. In the left panel, Cluster 1 (blue) corresponds to canonical BE epithelium. Cluster 3 (green) defines a spatially confined epithelial patch with elevated CEACAM5, CEACAM6, and CLDN4, suggestive of early neoplastic transformation. Cluster 2 (orange) localizes to the columnar–squamous interface and expresses squamous markers (KRT5, KRT13) along with inflammatory genes (S100A8, S100A9), consistent with a reactive squamous phenotype. The region of high-grade dysplasia (HGD) was labeled. In the right panel, Spatial transcriptomics map of tissue compartments derived from morphology-aligned gene expression using Visium HD at 2 µm resolution. BE epithelium (light blue) expresses MUC2, TFF3, REG4, CDX2 consistent with intestinal metaplasia, while EAC regions (dark blue) are defined by expression of MKI67, SPINK1, EiRBB2, and CLDN4. Squamous epithelium (dark green) is defined by DSG3, KRT5, KRT14, and TP63. Stromal regions (light green) express ACTA2, PDGFRA, COL1A1, COL3A1, FAP, MMP2, and MMP9, indicating fibroblast activation and extracellular matrix remodeling.This case serves as a representative example demonstrating the spatial distribution of transcriptomic clusters within a BE sample containing.

**Extended Data Table 1 | Definitions of Barrett's esophagus phenotype groups and baseline demographics of esophageal adenocarcinoma cases stratified by phenotype**

| Pathological evidence of adjacent intestinal metaplasia | Endoscopic evidence of Barrett's oesophagus | Barrett's phenotype | N | Percentage |
|---|---|---|---|---|
| Negative | Negative | Negative | 880 | 28.40% |
| Positive | Positive | Positive | 357 | 11.50% |
| Positive | Negative | Positive | 279 | 9.00% |
| N/A | Positive | Positive | 277 | 8.90% |
| Positive | N/A | Positive | 175 | 5.60% |
| Negative | Positive | Positive | 147 | 4.70% |
| Negative | N/A | Unknown | 81 | 2.60% |
| N/A | Negative | Unknown | 243 | 7.80% |
| N/A | N/A | Unknown | 661 | 21.30% |
| | | Total | 3100 | 100% |

| Characteristic | BE+ve EAC (n=1,235) | BE-ve EAC (n=880) | BE(?) EAC (n=985) | Overall (n=3,100) |
|---|---|---|---|---|
| **Age at diagnosis; *years*** | | | | |
| Mean [SD] | 66.7 [9.2] | 65.5 [9.8] | 65.3 [10.2] | 65.9 [9.7] |
| Median [Q1, Q3] | 67.5 [61.4, 73.5] | 66.7 [59.0, 72.6] | 66.7 [58.6, 72.9] | 67.0 [59.8, 73.0] |
| **Age groups at diagnosis, n (%)** | | | | |
| < 50 years old | 67 (5.4) | 59 (6.7) | 75 (7.6) | 201 (6.5) |
| 50 - 59 years old | 193 (15.6) | 184 (20.9) | 213 (21.6) | 590 (19.0) |
| 60 - 69 years old | 497 (40.2) | 317 (36.0) | 333 (33.8) | 1147 (37.0) |
| 70+ years old | 478 (38.7) | 320 (36.4) | 364 (37.0) | 1162 (37.5) |
| **Gender, n (%)** | | | | |
| Female | 163 (13.2) | 158 (18.0) | 161 (16.3) | 482 (15.5) |
| Male | 1072 (86.8) | 722 (82.0) | 824 (83.7) | 2618 (84.5) |
| **Ethnicity, n (%)** | | | | |
| Other | 20 (1.71) | 11 (1.32) | 23 (2.42) | 54 (1.83) |
| White Caucasian | 1152 (98.3) | 822 (98.7) | 928 (97.6) | 2902 (98.2) |
| Missing | 63 (5.1) | 47 (5.3) | 34 (3.5) | 144 (4.6) |

A. The definition and split of patients classified by Barrett's esophagus positive, negative, and unknown phenotypes. B. Baseline demographics of esophageal adenocarcinoma (EAC) cases overall and according to tumour phenotype and including BE(?) cases.

**Extended Data Table 2 | Sensitivity and comparative analyses of baseline characteristics associated with esophageal adenocarcinoma (EAC) phenotype**

| Characteristic | Multiple imputation Fully adjusted model[a] OR (95% CI, p) | Complete case analysis BE+ve EAC n (%) | BE-ve EAC n (%) | Fully adjusted model[a] OR (95% CI, p) | Excluding BE surveillance EAC cases (n=214) BE+ve EAC n (%) | BE-ve EAC n (%) | Fully adjusted model[a] OR (95% CI, p) |
|---|---|---|---|---|---|---|---|
| **Age group at diagnosis** | | | | | | | |
| < 50 | 1.00 (Referent) | 28 (6.2) | 24 (7.5) | 1.00 (Referent) | 56 (5.1) | 59 (6.9) | 1.00 (Referent) |
| 50 – 59 | 1.13 (0.74-1.73, p=0.577) | 65 (14.4) | 58 (18.1) | 1.20 (0.61-2.39, p=0.598) | 174 (16.0) | 179 (20.8) | 1.02 (0.66-1.58, p=0.917) |
| 60 – 69 | 0.72 (0.48-1.07, p=0.107) | 168 (37.2) | 130 (40.5) | 1.02 (0.55-1.92, p=0.946) | 437 (40.1) | 309 (36.0) | 0.68 (0.45-1.02, p=0.064) |
| 70+ | 0.70 (0.47-1.04, p=0.077) | 191 (42.3) | 109 (34.0) | 0.69 (0.37-1.31, p=0.253) | 423 (38.8) | 312 (36.3) | 0.68 (0.45-1.02, p=0.064) |
| **Gender** | | | | | | | |
| Female | 1.00 (Referent) | 56 (12.4) | 58 (18.1) | 1.00 (Referent) | 144 (13.2) | 151 (17.6) | 1.00 (Referent) |
| Male | 0.67 (0.52-0.87, p=0.002) | 396 (87.6) | 263 (81.9) | 0.61 (0.40-0.93, p=0.022) | 946 (86.8) | 708 (82.4) | 0.68 (0.53-0.88, p=0.004) |
| **BMI group at baseline** | | | | | | | |
| Normal | 1.00 (Referent) | 126 (27.9) | 99 (30.8) | 1.00 (Referent) | 245 (22.5) | 248 (28.9) | 1.00 (Referent) |
| Overweight | 0.89 (0.70-1.13, p=0.359) | 191 (42.3) | 143 (44.5) | 1.10 (0.77-1.57, p=0.612) | 362 (33.2) | 307 (35.7) | 0.91 (0.72-1.16, p=0.445) |
| Obese | 0.57 (0.44-0.75, p<0.001) | 135 (29.9) | 79 (24.6) | 0.78 (0.52-1.16, p=0.222) | 258 (23.7) | 154 (17.9) | 0.63 (0.48-0.83, p=0.001) |
| Missing | - | - | - | - | 225 (20.6) | 150 (17.5) | 0.93 (0.67-1.29, p=0.662) |
| **Cigarette smoking status** | | | | | | | |
| Never | 1.00 (Referent) | 155 (34.3) | 89 (27.7) | 1.00 (Referent) | 398 (36.5) | 291 (33.9) | 1.00 (Referent) |
| Ever | 1.21 (0.99-1.47, p=0.060) | 297 (65.7) | 232 (72.3) | 1.34 (0.97-1.86, p=0.082) | 574 (52.7) | 518 (60.3) | 1.29 (1.05-1.58, p=0.016) |
| Missing | - | - | - | - | 118 (10.8) | 50 (5.8) | 0.48 (0.30-0.76, p=0.002) |
| **Aspirin/NSAID use** | | | | | | | |
| Never | 1.00 (Referent) | 208 (46.0) | 129 (40.2) | 1.00 (Referent) | 235 (21.6) | 159 (18.5) | 1.00 (Referent) |
| Ever | 1.36 (1.04-1.77, p=0.023) | 244 (54.0) | 192 (59.8) | 1.45 (1.08-2.00, p=0.015) | 277 (25.4) | 235 (27.4) | 1.40 (1.06-1.85, p=0.017) |
| Missing | - | - | - | - | 578 (53.0) | 465 (54.1) | 1.39 (1.07-1.81, p=0.014) |
| **Heartburn symptoms status** | | | | | | | |
| Absent | 1.00 (Referent) | 70 (15.5) | 73 (22.7) | 1.00 (Referent) | 145 (13.3) | 159 (18.5) | 1.00 (Referent) |
| Present | 0.62 (0.47-0.81, p=0.001) | 382 (84.5) | 248 (77.3) | 0.56 (0.38-0.83, p=0.004) | 684 (62.8) | 524 (61.0) | 0.69 (0.53-0.89, p=0.005) |
| Missing | - | - | - | - | 261 (23.9) | 176 (20.5) | 0.79 (0.55-1.12, p=0.182) |
| **TNM** | | | | | | | |
| I | 1.00 (Referent) | 132 (29.2) | 44 (13.7) | 1.00 (Referent) | 232 (21.3) | 80 (9.3) | 1.00 (Referent) |
| II | 2.40 (1.78-3.25, p<0.001) | 110 (24.3) | 85 (26.5) | 2.33 (1.48-3.69, p<0.001) | 247 (22.5) | 197 (22.9) | 2.31 (1.68-3.20, p<0.001) |
| III | 2.92 (2.22-3.84, p<0.001) | 200 (44.2) | 186 (57.9) | 2.68 (1.80-4.06, p<0.001) | 459 (42.1) | 440 (51.2) | 2.61 (1.96-3.50, p<0.001) |
| IV | 3.16 (1.73-5.78, p<0.001) | 10 (2.2) | 6 (1.9) | 1.92 (0.60-5.69, p=0.250) | 25 (2.3) | 26 (3.0) | 2.84 (1.53-5.28, p=0.001) |
| Missing | - | - | - | - | 129 (11.8) | 116 (13.5) | 2.63 (1.83-3.80, p<0.001) |

[a]adjusted for all covariates. *all p-values and 95% CIs are two-sided without adjustment for multiple comparisons significant and ≤0.05.

| Characteristic | BE(?) EAC n (%) | BE+ve EAC n (%) | Univariable model OR (95% CI, p) | Minimally adjusted model[a] OR (95% CI, p) | Fully adjusted model[b] OR (95% CI, p) | Fully adjusted model excluding heartburn OR (95% CI, p) |
|---|---|---|---|---|---|---|
| **Age group at diagnosis** | | | | | | |
| < 50 | 75 (7.6) | 67 (5.4) | 1.00 (Referent) | 1.00 (Referent) | 1.00 (Referent) | 1.00 (Referent) |
| 50 – 59 | 213 (21.6) | 193 (15.6) | 1.01 (0.69-1.49, p=0.942) | 0.99 (0.68-1.46, p=0.965) | 0.98 (0.66-1.46, p=0.929) | 0.99 (0.67-1.47, p=0.969) |
| 60 – 69 | 333 (33.8) | 497 (40.2) | 1.67 (1.17-2.39, p=0.005) | 1.63 (1.14-2.33, p=0.008) | 1.64 (1.13-2.38, p=0.009) | 1.64 (1.14-2.38, p=0.008) |
| 70+ | 364 (37.0) | 478 (38.7) | 1.47 (1.03-2.10, p=0.034) | 1.44 (1.01-2.06, p=0.045) | 1.46 (1.01-2.11, p=0.047) | 1.45 (1.00-2.10, p=0.049) |
| **Gender** | | | | | | |
| Female | 161 (16.3) | 163 (13.2) | 1.00 (Referent) | 1.00 (Referent) | 1.00 (Referent) | 1.00 (Referent) |
| Male | 824 (83.7) | 1072 (86.8) | 1.29 (1.01-1.63, p=0.037) | 1.25 (0.99-1.59, p=0.063) | 1.32 (1.03-1.70, p=0.026) | 1.30 (1.02-1.67, p=0.036) |
| **BMI group at baseline** | | | | | | |
| Normal | 248 (25.2) | 268 (21.7) | 1.00 (Referent) | 1.00 (Referent) | 1.00 (Referent) | 1.00 (Referent) |
| Overweight | 307 (31.2) | 422 (34.2) | 1.27 (1.01-1.60, p=0.038) | 1.25 (1.00-1.58, p=0.052) | 1.14 (0.90-1.44, p=0.282) | 1.16 (0.91-1.46, p=0.227) |
| Obese | 195 (19.8) | 304 (24.6) | 1.44 (1.12-1.85, p=0.004) | 1.49 (1.16-1.92, p=0.002) | 1.34 (1.03-1.74, p=0.028) | 1.36 (1.05-1.76, p=0.021) |
| Missing | 235 (23.9) | 241 (19.5) | 0.95 (0.74-1.22, p=0.681) | 0.96 (0.75-1.24, p=0.763) | 0.90 (0.66-1.24, p=0.522) | 0.89 (0.65-1.22, p=0.469) |
| **Cigarette smoking status** | | | | | | |
| Never | 304 (30.9) | 452 (36.6) | 1.00 (Referent) | 1.00 (Referent) | 1.00 (Referent) | 1.00 (Referent) |
| Ever | 562 (57.1) | 665 (53.8) | 0.80 (0.66-0.96, p=0.015) | 0.79 (0.65-0.95, p=0.011) | 0.78 (0.64-0.95, p=0.014) | 0.79 (0.65-0.96, p=0.019) |
| Missing | 119 (12.1) | 118 (9.6) | 0.67 (0.50-0.89, p=0.007) | 0.67 (0.50-0.90, p=0.007) | 0.88 (0.59-1.32, p=0.531) | 0.82 (0.57-1.20, p=0.310) |
| **Aspirin/NSAID use** | | | | | | |
| Never | 201 (20.4) | 257 (20.8) | 1.00 (Referent) | 1.00 (Referent) | 1.00 (Referent) | 1.00 (Referent) |
| Ever | 263 (26.7) | 328 (26.6) | 0.98 (0.76-1.25, p=0.842) | 0.94 (0.73-1.20, p=0.613) | 0.92 (0.71-1.19, p=0.521) | 0.94 (0.73-1.21, p=0.635) |
| Missing | 521 (52.9) | 650 (52.6) | 0.98 (0.78-1.21, p=0.825) | 0.96 (0.77-1.20, p=0.716) | 1.03 (0.80-1.31, p=0.835) | 1.00 (0.79-1.27, p=0.976) |
| **Heartburn symptoms status** | | | | | | |
| Absent | 144 (14.6) | 150 (12.1) | 1.00 (Referent) | 1.00 (Referent) | 1.00 (Referent) | - |
| Present | 599 (60.8) | 818 (66.2) | 1.31 (1.02-1.69, p=0.035) | 1.34 (1.04-1.72, p=0.025) | 1.29 (0.99-1.67, p=0.060) | - |
| Missing | 242 (24.6) | 267 (21.6) | 1.06 (0.79-1.41, p=0.695) | 1.07 (0.80-1.43, p=0.644) | 1.08 (0.76-1.54, p=0.670) | - |
| **TNM** | | | | | | |
| I | 116 (11.8) | 294 (23.8) | 1.00 (Referent) | 1.00 (Referent) | 1.00 (Referent) | 1.00 (Referent) |
| II | 192 (19.5) | 279 (22.6) | 0.57 (0.43-0.76, p<0.001) | 0.56 (0.42-0.75, p<0.001) | 0.57 (0.43-0.76, p<0.001) | 0.58 (0.43-0.77, p<0.001) |
| III | 424 (43.0) | 481 (38.9) | 0.45 (0.35-0.57, p<0.001) | 0.45 (0.35-0.58, p<0.001) | 0.48 (0.37-0.62, p<0.001) | 0.47 (0.37-0.61, p<0.001) |
| IV | 18 (1.8) | 26 (2.1) | 0.57 (0.30-1.09, p=0.084) | 0.57 (0.30-1.09, p=0.089) | 0.58 (0.31-1.12, p=0.101) | 0.57 (0.30-1.10, p=0.090) |
| Missing | 235 (23.9) | 155 (12.6) | 0.26 (0.19-0.35, p<0.001) | 0.26 (0.19-0.34, p<0.001) | 0.27 (0.20-0.36, p<0.001) | 0.26 (0.20-0.36, p<0.001) |

[a]adjusted for age group at diagnosis and gender.
[b]adjusted for all covariates. *all p-values and 95% CIs are two-sided without adjustment for multiple comparisons significant and ≤0.05.

| Characteristic | BE(?) EAC n (%) | BE-ve EAC n (%) | Univariable model OR (95% CI, p) | Minimally adjusted model[a] OR (95% CI, p) | Fully adjusted model[b] OR (95% CI, p) | Fully adjusted model excluding heartburn OR (95% CI, p) |
|---|---|---|---|---|---|---|
| **Age group at diagnosis** | | | | | | |
| < 50 | 75 (7.6) | 59 (6.7) | 1.00 (Referent) | 1.00 (Referent) | 1.00 (Referent) | 1.00 (Referent) |
| 50 – 59 | 213 (21.6) | 184 (20.9) | 1.10 (0.74-1.63, p=0.642) | 1.11 (0.75-1.65, p=0.612) | 1.15 (0.77-1.73, p=0.494) | 1.14 (0.76-1.71, p=0.529) |
| 60 – 69 | 333 (33.8) | 317 (36.0) | 1.21 (0.83-1.76, p=0.318) | 1.22 (0.84-1.78, p=0.295) | 1.24 (0.84-1.82, p=0.279) | 1.23 (0.84-1.81, p=0.296) |
| 70+ | 364 (37.0) | 320 (36.4) | 1.12 (0.77-1.63, p=0.559) | 1.13 (0.78-1.64, p=0.530) | 1.14 (0.78-1.68, p=0.497) | 1.14 (0.78-1.67, p=0.507) |
| **Gender** | | | | | | |
| Female | 161 (16.3) | 158 (18.0) | 1.00 (Referent) | 1.00 (Referent) | 1.00 (Referent) | 1.00 (Referent) |
| Male | 824 (83.7) | 722 (82.0) | 0.89 (0.70-1.14, p=0.357) | 0.89 (0.70-1.13, p=0.332) | 0.87 (0.68-1.12, p=0.277) | 0.88 (0.69-1.13, p=0.322) |
| **BMI group at baseline** | | | | | | |
| Normal | 248 (25.2) | 255 (29.0) | 1.00 (Referent) | 1.00 (Referent) | 1.00 (Referent) | 1.00 (Referent) |
| Overweight | 307 (31.2) | 317 (36.0) | 1.00 (0.79-1.27, p=0.972) | 1.01 (0.80-1.28, p=0.928) | 1.00 (0.79-1.27, p=0.995) | 1.00 (0.78-1.27, p=0.970) |
| Obese | 195 (19.8) | 155 (17.6) | 0.77 (0.59-1.02, p=0.066) | 0.78 (0.59-1.03, p=0.075) | 0.79 (0.60-1.05, p=0.101) | 0.79 (0.60-1.04, p=0.098) |
| Missing | 235 (23.9) | 153 (17.4) | 0.63 (0.48-0.83, p=0.001) | 0.63 (0.48-0.83, p=0.001) | 0.78 (0.56-1.09, p=0.143) | 0.79 (0.57-1.09, p=0.153) |
| **Cigarette smoking status** | | | | | | |
| Never | 304 (30.9) | 298 (33.9) | 1.00 (Referent) | 1.00 (Referent) | 1.00 (Referent) | 1.00 (Referent) |
| Ever | 562 (57.1) | 532 (60.5) | 0.97 (0.79-1.18, p=0.731) | 0.97 (0.79-1.18, p=0.739) | 0.98 (0.79-1.21, p=0.839) | 0.97 (0.79-1.20, p=0.790) |
| Missing | 119 (12.1) | 50 (5.7) | 0.43 (0.30-0.62, p<0.001) | 0.43 (0.29-0.61, p<0.001) | 0.44 (0.27-0.69, p=0.001) | 0.43 (0.28-0.66, p<0.001) |
| **Aspirin/NSAID use** | | | | | | |
| Never | 201 (20.4) | 161 (18.3) | 1.00 (Referent) | 1.00 (Referent) | 1.00 (Referent) | 1.00 (Referent) |
| Ever | 263 (26.7) | 243 (27.6) | 1.15 (0.88-1.51, p=0.301) | 1.15 (0.88-1.51, p=0.307) | 1.18 (0.89-1.56, p=0.247) | 1.14 (0.86-1.50, p=0.354) |
| Missing | 521 (52.9) | 476 (54.1) | 1.14 (0.90-1.45, p=0.286) | 1.13 (0.89-1.45, p=0.309) | 1.33 (1.02-1.74, p=0.036) | 1.31 (1.02-1.70, p=0.037) |
| **Heartburn symptoms status** | | | | | | |
| Absent | 144 (14.6) | 160 (18.2) | 1.00 (Referent) | 1.00 (Referent) | 1.00 (Referent) | - |
| Present | 599 (60.8) | 543 (61.7) | 0.82 (0.63-1.05, p=0.115) | 0.81 (0.63-1.04, p=0.101) | 0.80 (0.61-1.03, p=0.085) | - |
| Missing | 242 (24.6) | 177 (20.1) | 0.66 (0.49-0.89, p=0.006) | 0.65 (0.48-0.88, p=0.005) | 0.83 (0.58-1.19, p=0.322) | - |
| **TNM** | | | | | | |
| I | 116 (11.8) | 87 (9.9) | 1.00 (Referent) | 1.00 (Referent) | 1.00 (Referent) | 1.00 (Referent) |
| II | 192 (19.5) | 200 (22.7) | 1.39 (0.99-1.96, p=0.059) | 1.40 (0.99-1.97, p=0.056) | 1.39 (0.98-1.96, p=0.065) | 1.37 (0.97-1.95, p=0.071) |
| III | 424 (43.0) | 447 (50.8) | 1.41 (1.03-1.92, p=0.030) | 1.42 (1.04-1.94, p=0.026) | 1.42 (1.03-1.94, p=0.030) | 1.41 (1.03-1.94, p=0.031) |
| IV | 18 (1.8) | 26 (3.0) | 1.93 (1.00-3.78, p=0.052) | 1.95 (1.01-3.84, p=0.048) | 1.94 (1.00-3.86, p=0.054) | 1.94 (1.00-3.85, p=0.054) |
| Missing | 235 (23.9) | 120 (13.6) | 0.68 (0.48-0.97, p=0.034) | 0.69 (0.48-0.98, p=0.038) | 0.67 (0.47-0.96, p=0.029) | 0.67 (0.47-0.96, p=0.027) |

[a]adjusted for age group at diagnosis and gender.
[b]adjusted for all covariates. *all p-values and 95% CIs are two-sided without adjustment for multiple comparisons significant and ≤0.05.

A. Sensitivity analyses of the adjusted association of baseline characteristics with risk of Barrett's esophagus (BE) negative (BE-ve) esophageal adenocarcinoma (EAC) compared to BE positive (BE+ve) EAC cases using multiple imputation, complete case analysis and exclusion of EAC cases with a history of undergoing BE surveillance. B. Associations of baseline characteristics with risk of BE+ve EAC phenotype compared to BE(?) EAC cases. C. Associations of baseline characteristics with risk of BE-ve EAC phenotype compared to BE(?) EAC cases.

**Extended Data Table 3 | Baseline characteristics of the esophageal adenocarcinoma (EAC) cases with whole-genome sequencing, overall and according to phenotype**

| Characteristic | BE+ve EAC (n=252) | BE-ve EAC (n=183) | BE(?) EAC (n=275) | Overall (n=710) |
|---|---|---|---|---|
| **Age at diagnosis; *years*** | | | | |
| Mean [SD] | 67.2 [9.2] | 65.6 [9.5] | 67.4 [10.0] | 66.9 [9.6] |
| Median [Q1, Q3] | 67.7 [61.4, 74.5] | 66.9 [59.3, 72.7] | 68.0 [60.4, 75.4] | 67.6 [60.4, 74.0] |
| **Age group at diagnosis, n (%)** | | | | |
| < 50 years old | 10 (4.0) | 10 (5.5) | 14 (5.1) | 34 (4.8) |
| 50 - 59 years old | 44 (17.5) | 39 (21.3) | 53 (19.3) | 136 (19.2) |
| 60 - 69 years old | 94 (37.3) | 69 (37.7) | 86 (31.3) | 249 (35.1) |
| 70+ years old | 104 (41.3) | 65 (35.5) | 122 (44.3) | 288 (40.6) |
| **Gender, n (%)** | | | | |
| Female | 33 (13.1) | 27 (14.8) | 47 (17.1) | 107 (15.0) |
| Male | 219 (86.9) | 156 (85.2) | 228 (82.9) | 603 (84.9) |
| **Ethnicity, n (%)** | | | | |
| White | 241 (95.6) | 175 (95.6) | 266 (96.7) | 682 (96.1) |
| Other | 0 (0) | 1 (0.5) | 2 (0.7) | 3 (0.4) |
| Missing | 11 (4.4) | 7 (3.8) | 7 (2.5) | 25 (3.5) |
| **BMI group at diagnosis, n (%)** | | | | |
| Underweight | 0 (0) | 2 (1.1) | 10 (3.6) | 12 (1.7) |
| Normal | 59 (23.4) | 47 (25.7) | 59 (21.5) | 165 (23.2) |
| Overweight | 77 (30.6) | 70 (38.3) | 81 (29.5) | 228 (32.1) |
| Obese | 70 (27.8) | 39 (21.3) | 55 (20.0) | 164 (23.1) |
| Missing | 46 (18.3) | 25 (13.7) | 70 (25.5) | 141 (19.9) |
| **Cigarette smoking status, n (%)** | | | | |
| Never | 88 (34.9) | 60 (32.8) | 86 (31.3) | 234 (33.0) |
| Ever | 151 (59.9) | 117 (63.9) | 169 (61.5) | 437 (61.5) |
| Missing | 13 (5.2) | 6 (3.3) | 20 (7.3) | 39 (5.5) |
| **Aspirin/NSAID use, n (%)** | | | | |
| Never | 56 (22.2) | 38 (20.8) | 46 (16.7) | 140 (19.7) |
| Ever | 76 (30.2) | 47 (25.7) | 59 (21.5) | 182 (25.6) |
| Missing | 120 (47.6) | 98 (53.6) | 170 (61.8) | 388 (54.6) |
| **Heartburn symptoms status, n (%)** | | | | |
| Absent | 35 (13.9) | 34 (18.6) | 47 (17.1) | 116 (16.3) |
| Present | 169 (67.1) | 107 (58.5) | 159 (57.8) | 435 (61.3) |
| Missing | 48 (19.0) | 42 (23.0) | 69 (25.1) | 159 (22.4) |
| **Total alcohol intake/week; *units*** | | | | |
| Mean [SD] | 22.8 [32.6] | 22.1 [17.1] | 29.2 [51.4] | 24.7 [36.7] |
| Median [Q1, Q3] | 15.0 [4.75, 22.5] | 20.0 [9.00, 28.0] | 18.0 [6.00, 35.0] | 18.0 [7.00, 30.0] |
| Missing | 200 (79.4) | 138 (75.4) | 226 (82.2) | 564 (79.4) |
| **TNM, n (%)** | | | | |
| I | 62 (24.6) | 21 (11.5) | 12 (4.4) | 95 (13.4) |
| II | 54 (21.4) | 31 (16.9) | 23 (8.4) | 108 (15.2) |
| III | 98 (38.9) | 97 (53.0) | 66 (24.0) | 261 (36.8) |
| IV | 4 (1.6) | 10 (5.5) | 1 (0.4) | 15 (2.1) |
| Missing | 34 (13.5) | 24 (13.1) | 173 (62.9) | 231 (32.5) |

**Extended Data Table 4 | T stage distribution across different phenotypes of esophageal adenocarcinoma (EAC) in the full cohort and the surveillance cohort**

| | Full cohort (n=3,100) | | | Surveillance cohort (n=214) | | |
|---|---|---|---|---|---|---|
| | BE+ve (n=1,235) | BE-ve (n=880) | BE? ve (n=985) | BE+ve (n=145) | BE-ve (n=21) | BE? ve (n=48) |
| T1 | 25.6% (316) | 9.9% (87) | 12.7% (125) | 54.5% (79) | 23.8% (5) | 37.5% (18) |
| T2 | 13.0% (161) | 10.3% (91) | 11.4% (112) | 13.1% (19) | 9.5% (2) | 8.3% (4) |
| T3 | 48.4% (598) | 61.6% (542) | 49.1% (484) | 20% (29) | 47.6% (10) | 25% (12) |
| T4 | 5.3% (65) | 9.1% (80) | 8.0% (79) | 0.7% (1) | 4.8% (1) | 4.2% (2) |
| Missing | 7.7% (95) | 9.1% (80) | 18.8% (185) | 11.7% (17) | 14.3% (3) | 25% (12) |

**Extended Data Table 5 | Functional gene panels for tissue identity, immune profiling, and quality control**

| Panel Category | Representative Genes |
|---|---|
| **Housekeeping Genes** | |
| | *ACTB*[1], *GAPDH*[1], *RPLP0*[1], *RPS18*[1], *EEF1A1*[1], *TUBB*[1], *B2M*[1], *PPIA*[1], *UBC*[1], *COL1A1*[2], *ACTA2*[2], *VIM*[3] |
| **Tissue Classification Genes** | |
| Tumour | *MKI67*[4], *SPINK1*[5], *ERBB2*[6], *CLDN4*[7] |
| Barrett's | *MUC2*[8], *TFF3*[9], *REG4*[10], *CDX2*[11] |
| Stroma | *ACTA2*[2,12], *PDGFRA*[13], *COL1A1*[2,14], *COL3A1*[14], *FAP*[15], *MMP2*[16], *MMP9*[16] |
| Squamous | *DSG3*[17], *KRT5*[18], *KRT14*[19], *TP63*[20] |

[1]National Center for Biotechnology Information. *Gene.* National Library of Medicine, [2]Qi W et al. *Ann Transl Med* (2021), [3]Moinova HR et al. *Sci Transl Med* (2018), [4]Matani H et al. *Int J Radiat Oncol Biol Phys* (2021), [5]Khorfan K et al. *J Clin Oncol* (2019), Nancarrow DJ et al. *PloS One* (2011), Räsänen K et al. *Clin Chem.* (2016), [6]Dahlberg P et al. *Ann Thorac Surg* (2004), [7]Ebbing EA et al. *Gastroenterology* (2017), Liu H et al, *Cancer Gene Ther* (2021), Hashimoto et al. *Cancers* (2022) [8]Steininger H et al. *Pathol Res Pract* (2005), [9]Peitz U et al. *Peptides* (2004), [10]Busslinger GA, et al. *Proc Natl Acad Sci U.S.A* (2021), [11]Stairs DB, et al. *PloS One* (2008), [12]Park S et al. *Clin Cancer Res* (2023), [13]Yoshida K et al. *J Cancer Res Clin Oncol* (1993), Kumar N, et al. *Nat Commun* (2024) [14]Chen Y, Zhu et al. *Cancer Cell* (2023) , [15]Ha SY et al. *PloS One* (2014) , [16] Etoh T et al. *Gut* (2000) [17]Chen YJ, et al. *PloS One* (2013) [18]Jiang M et al. *Nature* (2017), [19]Su H et al. *Cancer Res* (2003), [20]Jiang YY et al. *Gastroenterology* (2020)

Curated list of genes selected through comprehensive literature review to enable spatial transcriptomics identification of tissue compartments, immune cell types, and to distinguish Barrett's esophagus from esophageal adenocarcinoma in biopsy samples.

# Reporting Summary

## Statistics

For all statistical analyses, confirm that the following items are present in the figure legend, table legend, main text, or Methods section.

| n/a | Confirmed | |
|---|---|---|
| ☐ | ☒ | The exact sample size (*n*) for each experimental group/condition, given as a discrete number and unit of measurement |
| ☐ | ☒ | A statement on whether measurements were taken from distinct samples or whether the same sample was measured repeatedly |
| ☐ | ☒ | The statistical test(s) used AND whether they are one- or two-sided *Only common tests should be described solely by name; describe more complex techniques in the Methods section.* |
| ☐ | ☒ | A description of all covariates tested |
| ☐ | ☒ | A description of any assumptions or corrections, such as tests of normality and adjustment for multiple comparisons |
| ☐ | ☒ | A full description of the statistical parameters including central tendency (e.g. means) or other basic estimates (e.g. regression coefficient) AND variation (e.g. standard deviation) or associated estimates of uncertainty (e.g. confidence intervals) |
| ☐ | ☒ | For null hypothesis testing, the test statistic (e.g. *F*, *t*, *r*) with confidence intervals, effect sizes, degrees of freedom and *P* value noted *Give P values as exact values whenever suitable.* |
| ☒ | ☐ | For Bayesian analysis, information on the choice of priors and Markov chain Monte Carlo settings |
| ☐ | ☒ | For hierarchical and complex designs, identification of the appropriate level for tests and full reporting of outcomes |
| ☒ | ☐ | Estimates of effect sizes (e.g. Cohen's *d*, Pearson's *r*), indicating how they were calculated |

*Our web collection on statistics for biologists contains articles on many of the points above.*

## Software and code

Policy information about availability of computer code

| Data collection | n/a |
|---|---|
| Data analysis | https://github.com/fitzgerald-lab/EAC-Phenotypes - analysis pipeline, packages and versions are detailed here |

For manuscripts utilizing custom algorithms or software that are central to the research but not yet described in published literature, software must be made available to editors and reviewers. We strongly encourage code deposition in a community repository (e.g. GitHub). See the Nature Portfolio guidelines for submitting code & software for further information.

## Data

Policy information about availability of data

All manuscripts must include a data availability statement. This statement should provide the following information, where applicable:
- Accession codes, unique identifiers, or web links for publicly available datasets
- A description of any restrictions on data availability
- For clinical datasets or third party data, please ensure that the statement adheres to our policy

The data that support the findings of this work are available as follows: The data that support the findings of this work are available as follows: WGS data for samples sequenced by Illumina or the CRUK Cambridge Institute are available at the European Genome-phenome Archive (EGA) under accession number EGAD00001011191 . WGS data for samples sequenced by the Wellcome Sanger Institute are available at the EGA under accession number EGAD00001006083.

# Research involving human participants, their data, or biological material

Policy information about studies with human participants or human data. See also policy information about sex, gender (identity/presentation), and sexual orientation and race, ethnicity and racism.

| Reporting on sex and gender | Gender was determined based on self-reported questionnaire data |
|---|---|
| Reporting on race, ethnicity, or other socially relevant groupings | according to policy |
| Population characteristics | This is described in Table 1 and Extended Data Tables 1, 2 and 3 |
| Recruitment | Prospective, individual informed consent. Details are given in the manuscript "selection of cases" |
| Ethics oversight | East of England Ethics Committees |

Note that full information on the approval of the study protocol must also be provided in the manuscript.

# Field-specific reporting

Please select the one below that is the best fit for your research. If you are not sure, read the appropriate sections before making your selection.

☒ Life sciences    ☐ Behavioural & social sciences    ☐ Ecological, evolutionary & environmental sciences

For a reference copy of the document with all sections, see nature.com/documents/nr-reporting-summary-flat.pdf

# Life sciences study design

All studies must disclose on these points even when the disclosure is negative.

| Sample size | No statistical method was used to predetermine sample size. However, this study represents the largest clinical and genomic cohort to date, with the next largest cohorts having 54% fewer patients with clinical, and 31% and 49% fewer EAC and BE genomically sequenced patients, respectively. |
|---|---|
| Data exclusions | We removed samples/data of poor quality. Details are given in the manuscript |
| Replication | For the epidemiological and genomic data analysis, we used different methods (technical replicates) to provide evidence from different aspects. For staining, biological replicates were done with different markers. |
| Randomization | The experiments were not randomized. Randomization is not suitable as this is a real world dataset. |
| Blinding | The pathologist-investigators were blinded to biopsy results, the risk factor data (heartburn, smoking, and BMI). Other investigators were not blinded to the independent variables and phenotype outcome. |

# Reporting for specific materials, systems and methods

We require information from authors about some types of materials, experimental systems and methods used in many studies. Here, indicate whether each material, system or method listed is relevant to your study. If you are not sure if a list item applies to your research, read the appropriate section before selecting a response.

## Materials & experimental systems

| n/a | Involved in the study |
|---|---|
| ☐ | ☒ Antibodies |
| ☒ | ☐ Eukaryotic cell lines |
| ☒ | ☐ Palaeontology and archaeology |
| ☒ | ☐ Animals and other organisms |
| ☐ | ☒ Clinical data |
| ☒ | ☐ Dual use research of concern |
| ☒ | ☐ Plants |

## Methods

| n/a | Involved in the study |
|---|---|
| ☒ | ☐ ChIP-seq |
| ☒ | ☐ Flow cytometry |
| ☒ | ☐ MRI-based neuroimaging |

# Antibodies

| | |
|---|---|
| Antibodies used | We used TFF3 antibody from Invitrogen (0.26 ug/ml), and REG4 antibody from Abcam #ab255820 (0.26ug/ml). |
| Validation | According to the manufacturer, the TFF3 antibody (#MA5-42854, clone 7X6O2, RRID AB_2911995) was validated by IHC in HEK293 cells, and human colon carcinoma tissue (Thermofisher scientific datasheet), with published images on the product page. The REG4 antibody (#ab255820, clone EPR22810-327) was validated by the manufacturer on human colon tissue using IHC at 1/2000 (0.262 µg/ml) and positive staining observed, with published images on product page (Abcam datasheet). Specificity was confirmed by absence of signal in squamous epithelium of normal oesophageal biopsies, and strong expression in human colon tissue for both TFF3 and REG4 antibodies. |

# Clinical data

Policy information about clinical studies

All manuscripts should comply with the ICMJE guidelines for publication of clinical research and a completed CONSORT checklist must be included with all submissions.

| | |
|---|---|
| Clinical trial registration | UKCRNID-8880, REC 07/H0305/52 and 10/H0305/1 |
| Study protocol | OCCAMS study protocol available from study coordinators on request |
| Data collection | Prospective |
| Outcomes | epidemiological and genomic profiling correlated with presence or absence of Barretts Oesophagus |

# Plants

| | |
|---|---|
| Seed stocks | *Report on the source of all seed stocks or other plant material used. If applicable, state the seed stock centre and catalogue number. If plant specimens were collected from the field, describe the collection location, date and sampling procedures.* |
| Novel plant genotypes | *Describe the methods by which all novel plant genotypes were produced. This includes those generated by transgenic approaches, gene editing, chemical/radiation-based mutagenesis and hybridization. For transgenic lines, describe the transformation method, the number of independent lines analyzed and the generation upon which experiments were performed. For gene-edited lines, describe the editor used, the endogenous sequence targeted for editing, the targeting guide RNA sequence (if applicable) and how the editor was applied.* |
| Authentication | *Describe any authentication procedures for each seed stock used or novel genotype generated. Describe any experiments used to assess the effect of a mutation and, where applicable, how potential secondary effects (e.g. second site T-DNA insertions, mosiacism, off-target gene editing) were examined.* |

