## [Peer review file · Nature Medicine]

Integrated epidemiological and molecular data inform the relationship between precancer and cancer states of oesophageal adenocarcinoma

Corresponding Author: Professor Rebecca Fitzgerald

This manuscript has been previously reviewed at another journal. This document only contains information relating to versions considered at Nature Medicine.

Version 0:

Reviewer comments:

Reviewer #1

(Remarks to the Author)

This manuscript is one I have previously reviewed, and my assessment here focuses on the adequacy of the revisions and the current state of the work, acknowledging that the editors have full access to the earlier reviewer reports.

Overall, my prior support for this study is unchanged. I continue to view this as an important and clinically consequential manuscript that addresses, as definitively as available methods allow, the central question: Do all oesophageal adenocarcinomas (OACs) originate from Barrett's oesophagus? This is a critical issue with direct implications for patient management.

The key challenge the authors faced is that in OAC patients without visible Barrett's, there is no alternative pathology present to compare against. Therefore the strength of this work lies in its very large cohort and high-quality multi-omics dataset; if a distinct non-Barrett's precursor or divergent evolutionary trajectory existed, the authors' analyses are sufficiently sensitive that it should have been detected.

Multiple theories have been proposed regarding the cellular origins of Barrett's oesophagus, and by extension, possible origins of OAC. However, the strongest evidence to date supports a gastric epithelial origin, with subsequent intestinal metaplasia. While alternative sources such as oesophageal submucosal glands have been suggested, when identified these consistently maintain a clonal relationship with Barrett's, implying that intestinal metaplasia remains a necessary stage. The data presented here reinforce the concept that intestinal metaplasia is indeed an obligate precursor, even if earlier initiating events remain to be elucidated.

Regarding the other reviewers' comments, I find that the authors have addressed most points satisfactorily. However, a couple would still benefit from refinement:

Spatial transcriptomics (Reviewer 2):

The authors' written response is clear, but Figure 5B/C in its current form does not fully convey the lineage-supporting features described in the text. The Visium data would be more compelling if the key expression gradients underpinning the argument (particularly REG4 and TFF3) were visualised with greater clarity more directly comparable to the IHC. It is also unclear why frozen tissue was used when Visium HD performs well on FFPE with superior histologic morphology; higher-resolution images may help address this issue. I appreciate that re-doing this in FFPE is expensive and perhaps higher resolution images of the frozen samples will be sufficient.

Figure 5D: This panel still requires clarification. The authors should ensure it accurately reflects that the relevant molecular and lineage features are present across all disease contexts, without implying a temporal sequence that the data cannot support.

Reviewer 5's conceptual concern of whether we can be certain that all OACs arise from Barrett's, has been addressed appropriately. In my view, the authors rightly avoid absolute statements, but the totality of molecular, histologic, and genomic evidence strongly supports the conclusion that OACs originate from an intestinal metaplastic lineage, even when endoscopic Barrett's is absent.

In summary, I remain supportive of this manuscript. The authors clearly articulate the ongoing uncertainty around the origins of OAC and present rigorous data that substantially advance our understanding. I would only suggest the clarifications stated to the figures noted above be addressed.

Confidential Information Redacted

(Remarks on code availability)

Re: Nature Medicine paper NMED-A147205-T

Integrated epidemiological and molecular data yields insights into the relationship between precancer and cancer states of oesophageal adenocarcinoma

We thank the reviewer for their helpful comments and the journal editors' consideration of this revised manuscript. We have responded to all queries point-by-point below and indicated where changes have been made to the manuscript text. All Reviewer comments are in black and our responses in blue, the line numbers refer to the tracked changes version of the manuscript.

Reviewers' Comments:

Reviewer #1 (Remarks to the Author):

This manuscript is one I have previously reviewed, and my assessment here focuses on the adequacy of the revisions and the current state of the work, acknowledging that the editors have full access to the earlier reviewer reports.

Overall, my prior support for this study is unchanged. I continue to view this as an important and clinically consequential manuscript that addresses, as definitively as available methods allow, the central question: Do all oesophageal adenocarcinomas (OACs) originate from Barrett's oesophagus? This is a critical issue with direct implications for patient management.

The key challenge the authors faced is that in OAC patients without visible Barrett's, there is no alternative pathology present to compare against. Therefore the strength of this work lies in its very large cohort and high-quality multi-omics dataset; if a distinct non-Barrett's precursor or divergent evolutionary trajectory existed, the authors' analyses are sufficiently sensitive that it should have been detected.

Multiple theories have been proposed regarding the cellular origins of Barrett's oesophagus, and by extension, possible origins of OAC. However, the strongest evidence to date supports a gastric epithelial origin, with subsequent intestinal metaplasia. While alternative sources such as oesophageal submucosal glands have been suggested, when identified these consistently maintain a clonal relationship with Barrett's, implying that intestinal metaplasia remains a necessary stage. The data presented here reinforce the concept that intestinal metaplasia is indeed an obligate precursor, even if earlier initiating events remain to be elucidated.

Regarding the other reviewers' comments, I find that the authors have addressed most points satisfactorily. However, a couple would still benefit from refinement:

Spatial transcriptomics (Reviewer 2):

The authors' written response is clear, but Figure 5B/C in its current form does not fully convey the lineage-supporting features described in the text. The Visium data

would be more compelling if the key expression gradients underpinning the argument (particularly REG4 and TFF3) were visualised with greater clarity more directly comparable to the IHC. It is also unclear why frozen tissue was used when Visium HD performs well on FFPE with superior histologic morphology; higher-resolution images may help address this issue. I appreciate that re-doing this in FFPE is expensive and perhaps higher resolution images of the frozen samples will be sufficient.

Figure 5D: This panel still requires clarification. The authors should ensure it accurately reflects that the relevant molecular and lineage features are present across all disease contexts, without implying a temporal sequence that the data cannot support.

Reviewer 5's conceptual concern of whether we can be certain that all OACs arise from Barrett's, has been addressed appropriately. In my view, the authors rightly avoid absolute statements, but the totality of molecular, histologic, and genomic evidence strongly supports the conclusion that OACs originate from an intestinal metaplastic lineage, even when endoscopic Barrett's is absent.

In summary, I remain supportive of this manuscript. The authors clearly articulate the ongoing uncertainty around the origins of OAC and present rigorous data that substantially advance our understanding. I would only suggest the clarifications stated to the figures noted above be addressed.

Response: We sincerely thank the reviewer for their thoughtful re-evaluation of the manuscript and for their continued support. We appreciate the recognition of the clinical importance of this work and we agree with the reviewer's assessment that, taken together, the data strongly support intestinal metaplasia as an obligate precursor to oesophageal adenocarcinoma.

We have carefully addressed the remaining points raised relating to the spatial transcriptomics data, as detailed below.

Frozen tissue was selected over FFPE for Visium analysis because well-preserved fresh-frozen samples generally yield higher-quality RNA and improved transcriptomic signal (Jacobsen et al., 2023; Redmayne et al. 2019). We also confirmed that tissue morphology was well preserved in the frozen sections used for this study. This explanation has been added in the revised manuscript (Page 26, line 656-659).

We agree that clearer visualisation of lineage-supporting expression patterns is critical. In response, we have updated all spatial transcriptomics panels, and shared the raw figures in the link:

https://www.dropbox.com/scl/fi/xdo8heahq12vi6ojut9uf/spatial_raw_figures.zip?rlkey=6cpxu2vwdoqrse16xcvoitxd&st=uu32qjwi&dl=0. It is optional at the editor's discretion whether these can be included as supplementary figures.

Regarding Figure 5D. we have removed the directional arrow in the disease stage to avoid implying a temporal sequence, and added more molecular icons at various disease stages, to show they present in all disease context as suggested.

Full citations:

Jacobsen SB, Tfelt-Hansen J, Smerup MH, Andersen JD, Morling N. Comparison of whole transcriptome sequencing of fresh, frozen, and formalin-fixed, paraffin-embedded cardiac tissue. PLoS One 18, e0283159 (2023)

Redmayne N, Chavez SL. Optimizing Tissue Preservation for High-Resolution Confocal Imaging of Single-Molecule RNA-FISH. Curr Protoc Mol Biol 129, e107 (2019)

Editors' comments:

In addition to addressing the remaining points from the reviewer, please edit your manuscript to comply with our formatting guidelines for Articles, which are:

* Abstract: 200 words, unreferenced.

* Main text: 4000 words with subheadings for the Introduction, Results and Discussion. The discussion should also present the limitations of the study.

* References: up to 60 in the main text + 20 methods-only references

* Display items: up to 6 main display items (inclusive of figures and tables) and up to 10 Extended Data display items (inclusive of figures and tables). Extended Data are an integral part of the paper and only data that directly contribute to the main message should be presented.

* Online Methods: no word limit; please provide the methods consolidated in a single section at the end of the main text document

Response: Thanks for the guidance, we have modified the manuscript to meet all the requirements listed above. Please let us know if any further improvement needed.